# Making a Monkey out of Human Immunodeficiency Virus/Simian Immunodeficiency Virus Pathogenesis: Immune Cell Depletion Experiments as a Tool to Understand the Immune Correlates of Protection and Pathogenicity in HIV Infection

**DOI:** 10.3390/v16060972

**Published:** 2024-06-17

**Authors:** Jen Symmonds, Thaidra Gaufin, Cuiling Xu, Kevin D. Raehtz, Ruy M. Ribeiro, Ivona Pandrea, Cristian Apetrei

**Affiliations:** 1Department of Pathology, School of Medicine, University of Pittsburgh, Pittsburgh, PA 15261, USA; jes224@pitt.edu (J.S.); cux1@pitt.edu (C.X.); kdr26@pitt.edu (K.D.R.); pandrea@pitt.edu (I.P.); 2Department of Infectious Diseases and Microbiology, School of Public Health, University of Pittsburgh, Pittsburgh, PA 15261, USA; 3Tulane National Primate Research Center, Tulane University, Covington, LA 70433, USA; tgaufin@gmail.com; 4Division of Infectious Diseases, School of Medicine, University of Pittsburgh, Pittsburgh, PA 15261, USA; 5Theoretical Biology and Biophysics Group, Los Alamos National Laboratory, Los Alamos, NM 87545, USA

**Keywords:** human immunodeficiency virus, simian immunodeficiency virus, AIDS, non-human primates, T-lymphocytes, B-lymphocytes, natural killer cells, regulatory T-cells

## Abstract

Understanding the underlying mechanisms of HIV pathogenesis is critical for designing successful HIV vaccines and cure strategies. However, achieving this goal is complicated by the virus’s direct interactions with immune cells, the induction of persistent reservoirs in the immune system cells, and multiple strategies developed by the virus for immune evasion. Meanwhile, HIV and SIV infections induce a pandysfunction of the immune cell populations, making it difficult to untangle the various concurrent mechanisms of HIV pathogenesis. Over the years, one of the most successful approaches for dissecting the immune correlates of protection in HIV/SIV infection has been the in vivo depletion of various immune cell populations and assessment of the impact of these depletions on the outcome of infection in non-human primate models. Here, we present a detailed analysis of the strategies and results of manipulating SIV pathogenesis through in vivo depletions of key immune cells populations. Although each of these methods has its limitations, they have all contributed to our understanding of key pathogenic pathways in HIV/SIV infection.

## 1. Introduction

The World Health Organization estimates that >38.4 million people worldwide are infected with human immunodeficiency virus (HIV). In 2021 alone, approximately 650,000 deaths resulted from acquired immune deficiency syndrome (AIDS) [1]. The use of antiretroviral therapy (ART) has greatly extended the lifespan of patients, but ART is costly, needs to be lifelong, and is plagued by viral resistance [2]. An AIDS vaccine would be ideal, but, in spite of 40 years of research, remains elusive [3]. The general consensus is that the development of an effective HIV vaccine requires a better understanding of the HIV/SIV pathogenesis [4,5,6,7].

The continuum of partial successes [8], and even more so the continuum of failures of HIV vaccine clinical trials [9], point to a pressing need for understanding the correlates of immune protection against infection and disease progression [4,5,6,7]. Studies have shown that multiple types of immune responses against HIV cooperate to generate a polyfunctional response for the control of HIV-1 transmission and infection. However, a major limitation to studying the correlates of protection from HIV infection or disease protection is that the complexity of these responses does not permit their untangling in persons living with HIV (PWHs). Such “immune dissection” studies can be performed in animal models, particularly in non-human primates (NHPs), where selective in vivo depletion of cellular components of the immune system is feasible. Furthermore, both the dynamics of SIV infection and the impact of the in vivo dissection of immune responses can be closely monitored in NHP models through invasive sampling.

The most common animal model used for AIDS research is the SIV-infected rhesus macaque (RM) [10]. While recapitulating the major pathogenic features of HIV/AIDS, this approach does not use actual HIV-1, as NHPs cannot be infected with HIV-1 due to host species restrictions [11,12]. HIV-SIV chimeras, called simian–human immunodeficiency viruses (SHIVs), were produced to overcome this limitation [13,14,15,16], and these were instrumental for AIDS vaccine studies because they contain HIV envelope immunogens and can be used for evaluating anti-HIV antibodies in vivo [13,14,15,17]. Early SHIVs exhibited different biological properties from HIV or SIV [17,18]: most notably, they were very easy to control with different vaccine formulations [19,20,21]. Yet, those same vaccines failed to control either SIVmac [22] or HIV infection [4]. Recently, transmitted–founder SHIVs (i.e., those constructed using the backbone of the SIVmac and envelopes derived from viruses collected form acutely HIV-infected subjects) more closely reproduce the patterns of HIV/SIV infections [23,24].

SIV infections of RMs (*Macaca mulatta*), pigtailed macaques (PTMs, *Macaca nemestrina*), and cynomolgus macaques (CyMs, *Macaca fascicularis*) progress to simian AIDS in a variable time frame [10,25,26]. These SIV infections are characterized by (i) massive viral replication [27,28,29,30,31]; (ii) fulminant depletion of circulating CD4^+^ T-cells, which is even more pronounced at mucosal sites [32,33,34,35,36]; (iii) high levels of T-cell immune activation and systemic inflammation, the magnitude of which is predictive of disease progression [18,37]; and (iv) hypercoagulation, which is one of the best predictors for death in PWHs and SIV-infected macaques, even in ART-suppressed individuals [38,39]. The interactions among these factors cripple the immune system, leading to immunodeficiency and death [40,41].

SIVmac-infected RMs progress to AIDS significantly faster than PWHs (0.5–3 yrs. vs. 8–10 yrs., respectively) [42,43,44]. This accelerated pathogenicity is the main reason for which some researchers believe that SIVmac infections in RMs do not properly model PHWs [10]. However, understanding how the immune system in macaque models responds to SIV infection is a powerful tool for studies on vaccines, therapies, and cures [4,10,17,25].

Another approach for understanding the immune correlates of protection from disease progression is based on the study of the pathogenesis of SIV infections in their African NHP natural hosts. African NHP species, such as African green monkeys (AGMs, *Chlorocebus* genus), sooty mangabeys (SMs, *Cercocebus atys*), and mandrills (*Mandrillus sphinx*), are infected with species-specific viruses at high levels of prevalence and generally do not progress to AIDS upon natural or experimental infection [45,46,47,48,49,50,51,52,53,54,55,56,57,58,59,60,61,62,63,64,65]. These natural SIV infections are characterized by (i) highly active viral replication, with set-point viral loads (VLs) similar to or higher than those reported in pathogenic infections [45,46,47,48,49,50,51,52,53,54,55,56,57,58,59,60,61,62,63,64,65]; (ii) a transient depletion of peripheral CD4^+^ T-cells during acute infection, which rebound to pre-infection levels during chronic infection [52,53,54,55,60,61]; (iii) massive CD4^+^ T-cell depletion in the intestine, which is partially restored during chronic infection despite ongoing viral replication [50,56]; (iv) low levels of CCR5^+^ CD4^+^ T-cells in the blood and tissues [66]; (v) transient and moderate increases in T-cell immune activation during acute infection, with a return to baseline levels during chronic infection [52,54,56,58,67]; and (vi) controlled chronic systemic inflammation due to an exquisite ability to maintain a healthy mucosal barrier throughout the SIV infection [67,68], which thus prevents microbial translocation [56].

The interactions between these factors lead to an active SIV infection, which only rarely progresses to AIDS [69]. It is widely believed that through comparing and contrasting the key aspects of immune responses between pathogenic and nonpathogenic infections, we can understand the reasons for the lack of disease progression, thereby identifying indirectly the correlates of immune protection. Experimental studies in natural hosts are challenging, because colonies of African NHPs are scarce (and only for AGMs, SMs, and mandrills); most African NHPs are endangered (precluding invasive studies); their importation from Africa is difficult; and studies in the wild are virtually impossible [70,71].

Finally, multiple approaches are being developed to study HIV in animal models, including (i) the genetic engineering of HIV strains to minimally include SIV sequences that can overcome host restriction factors, such as the modified HIV-1 that includes the SIV *vif* gene and a 21-nucleotide region of SIV Gag CA (this virus can successfully infect PTMs [72]); and (ii) the development of “humanized mice”, engineered to support HIV-1 infection. This “humanized mice” model also has limitations due to the size and lifespan of mice compared to humans [73,74]. Moreover, mucosal colonization with human immune cells is still relatively limited [74]. As such, despite inherent limitations, NHPs infected with SIV are still the best animal model to accurately study the immune correlates of HIV pathogenesis [10].

## 2. The Immune Response to HIV/SIV Infection

One of the reasons HIV is so difficult to control is that one of its immune evasion mechanisms involves the destruction of pivotal elements of the immune system. On top of this, HIV does not induce any immune effectors that can clearly be associated with protection. These are the major reasons for the laborious experiments manipulating the pathogenesis of HIV in vivo through selective immune cell depletions.

### 2.1. Humoral Immune Response against HIV Infection

HIV infections should be prevented from infecting CD4^+^ T-cells by neutralizing antibodies (NAbs). The current licensed vaccines on the market are protective against various pathogens because they elicit both an antibody response and a CD8^+^ or a CD4^+^ T-cell response [75,76]. Generating a vaccine that is protective against an HIV infection would therefore need to elicit NAbs. However, in a natural HIV infection, the emergence of NAbs is delayed for 2–3 months [77]; by then, most of the effector memory CD4^+^ T-cells (the main target cell of the virus) are already depleted [13,18,32,34,35], signifying that NAbs appear too late in natural infection to impact disease progression. Moreover, NAbs typically fail to neutralize both current virus strains from a PWH or any new virus variants, being one step behind the virus’s evolution due to rapid changes accumulating in the virus *env* sequence and changes in glycosylation [77,78,79].

However, if generated prior to infection, NAbs have the potential to prevent infection. NAbs work by targeting envelope glycoproteins Gp120 and Gp41, particularly the V1, V2, and V3 loops and the receptor-binding site on Gp120 [79,80,81]. However, the heavy glycosylation of HIV and SIV gp120 makes it difficult to elicit NAbs [79,80,81]. During chronic infection, NAb responses “mature”, enabling them to be effective against a wider range of strains and interfere with the binding of Gp120 and CD4 [81]. These NAbs are coined “broadly NAbs” (bNAbs) and have been observed to recognize a wide range of V3 loops [82].

Non-neutralizing antibodies appear two to three weeks after infection [81]. They mediate Fc-mediated antibody effector functions, namely antibody-dependent cellular cytotoxicity (ADCC) and antibody-dependent cell-mediated virus inhibition (ADCVI), two mechanisms contributing to HIV/SIV control. As opposed to general antibody-dependent cellular phagocytosis (ADCP), whereby antibody-coated cells are engulfed by phagocytes such as macrophages, neutrophils, gamma-delta (γδ) T-cells, and natural killer cells (NKs), ADCC is a mechanism mediated classically by NK cells through the Fc receptors that recognize the Fc region of antibodies bound to virally infected cells, triggering the release of cytotoxic molecules. These molecules, which include perforin and granzyme, ultimately lead to the lysis and death of target cells.

Both long-term non-progressors and non-infected infants born to HIV-infected mothers show strong ADCC activity against HIV envelope glycoproteins [83]. Also, higher ADCC levels in SIVmac251-infected RMs correlate to lower VLs [83].

The main difference between ADCC and ADCVI is that the latter assay measures the inhibition of the virus using antibodies and effector cells rather than directly measuring the cell killing [84]. Both ADCC and ADCVI involve IgG antibodies [84]. Antibody activities against the principal immunodominant domain of Gp41 and the V3 loop, both of which are immunogenic and not hindered by antibody binding, occur frequently and have the ability to inhibit HIV replication in macrophages and immature dendritic cells [85].

Furthermore, antibody responses are connected through the complement system to the innate immune system. In the classical pathway, the complement system is activated by antibodies coating the pathogen or the infected cell when the Fc region binds to the C1 protein, triggering a complement cascade, which leads to the cleavage of C3. The C3 components then amplify the cascade further, eventually forming the membrane attack complex (MAC) on the surface of an infected cell. The MAC forms a pore in the lipid bilayer, disrupting cellular integrity and destroying the target cell [86].

Hessell et al. reported that complement activity was not important in the control of HIV replication in the presence of NAbs. However, the complement-dependent destruction of infected cells mediated by non-neutralizing antibodies was greater in sera from PWHs during acute infection and correlated with lower VLs, in contrast to chronic infection [87]. Experimental mutations of the Fc regions of the b12 NAb impairing either complement activation capability or both complement and FCγR binding resulted in modified NAbs that could still bind and neutralize the virus [88]. More RMs were protected from SHIV infection when treated with antibodies deficient in complement activation compared to those treated with antibodies with intact complement function [88].

Multiple vaccine trials have identified multiple humoral correlates of protection against HIV [89], the most important of which being the levels of IgG antibodies targeting the variable V1/V2 regions of the HIV-1 envelope glycoprotein, which are associated with a decreased risk of HIV infection [90,91,92]. Conversely, plasma Env IgA to specific envelope glycoproteins was associated with an increased risk [90,93]. There is a dysfunction of IgA-producing B-cells and plasma cells during HIV/SIV infection. The loss of IgA in plasma correlates with the loss of IgA in intestine during HIV/SIV infection. Hypergammaglobulinemia is associated with chronic SIV pathogenic infection but not with nonprogression in natural hosts [94,95,96,97]. The V2 loop is included in the CCR5 receptor-binding site and contains the α4β7 mucosal homing integrin-binding site [98,99]. This relationship may be significant, as mAbs against α4β7 protect against transmission. Furthermore, administration of the anti-α4β7 antibody to SIV-infected RMs undergoing ART improved infection outcomes, CD4^+^ T-cell preservation, and virus control [100,101]. A detailed analysis of the B-cell repertoire and V2 targeting with mAbs demonstrated that V2-specific antibodies mediate ADCC, neutralization, virus capture, and antibody-mediated phagocytosis [102,103,104,105]. Anti-Env IgG3 responses have also been found to correlate with decreased HIV-1 risk [92] and be associated with Fc-mediated antibody function [92]. This is not surprising, as multiple HIV-1 bNAbs are IgG3 [106,107]. IgG3-mediated phagocytosis of HIV-1 virions could be a mechanism by which IgG1 and IgG3 non-bNAbs contributed to the RV144 trial’s efficacy [89].

Complex antibody Fc–FcR interactions contribute to protective immunity, such as the vaccine-induced ADCC response [104], as well as antibody synergy [108], additivity [109], and interference [93] for virion capture. Note, however, that the immune correlates of humoral protection reported in the RV144 trial to be associated with a decreased risk of HIV-1 acquisition were not demonstrated to have a mechanistical basis either ex vivo or in other vaccine trials [89]. Moreover, some of the “correlates of protection” in the RV144 trial were also observed to occur in a failed trial, VAX003 [89]. As such, further investigations in animal models are needed to further dissect the correlates of vaccine protection.

### 2.2. T-Cell Immune Response during HIV Infection

Cellular immune responses belong to the adaptive arm of the immune system and are composed mainly of CD4^+^ T-cells (with the key subsets of T-follicular helper cells (Tfhs), Th17, and regulatory T-cells (Tregs)) and CD8^+^ T-cells. CD8^+^ T-cells include cytotoxic T-lymphocytes (CTLs), which enlist the help of CD4^+^ T-helper cells [110] to destroy cells infected with pathogens. The Th17 cells control inflammation at mucosal sites [111], while Tregs suppress the activation of the immune system [112] and mediate immune activation. Due to the failure of antibody-based vaccine trials [113] and other research showing the importance of CD8^+^ T-cells in modulating HIV replication [114,115,116,117,118], prior to the moderate success of the Thai trial, the variability of HIV Env, the difficulty inducing bNAbs, and the potential to attack conserved regions of HIV made CD8^+^ T-cell-based vaccines attractive. Therefore, CTLs have been proposed to be the best way to control viral replication, CD4^+^ T-cell loss, and disease progression in PWHs [4,77]. Nevertheless, the first and only CTL-based vaccine trial carried out thus far failed to control HIV infection [4,5].

#### 2.2.1. CD8^+^ T-Cells

CD8^+^ T-cells with CTL activity are important for the clearance of a number of virus-infected cells, such as Sendai, vaccinia, herpes, RSV, influenza, or cytomegalovirus [119,120]. Initial studies in PWHs have demonstrated the importance of the early appearance of CTL precursors in the blood [116], and established a clear inverse correlation between the levels of HIV-specific CTLs and the VLs, particularly during acute infection [114,116]. Studies in NHPs have also shown the importance of CD8^+^ T-cells in controlling viral replication. In RMs containing the MHC class I molecule Mamu-A*01, acute SIVmac infection is partially controlled (i.e., the VLs partially decline) coincidental with the appearance of virus-specific CTLs [117]. Also, CD8^+^ cell depletion resulted in a lack of control of virus replication, with rebounds of plasma VLs during both acute and chronic SIV infections in RMs [115,118]. Moreover, a trend towards lower VLs was reported for SMs with increased CTL activity against HIV Gag or Env during chronic SIVsmm infection [121].

HIV escapes CTL recognition by undergoing mutations in Gag, with the majority of substitutions accumulating at or near the epitopes recognized by CTLs [122]. Furthermore, when CTLs specific for the HIV Nef protein were infused to PWHs, viruses with Nef epitope deletions accumulated rapidly [123]. Because the virus can escape CTL detection, it is likely that CTLs help control viral replication [124].

The Nairobi Commercial Sex Worker cohort from Kenya comprises a group of women heavily exposed to HIV, but not infected (after being tested via serology or PCR) [125]. These women exhibit specific and different CTL responses to HIV than typical HIV-infected women, often displaying CTL responses against peptides not recognized in other infected women [125,126]. However, even though the CTL responses in these highly exposed but non-infected women are narrowly directed and lower in magnitude compared to HIV-infected women, the recognition of certain HIV epitopes by CTLs may provide a safeguard from infection [126,127]. Finally, HIV elite controllers (ECs) (i.e., individuals with the ability to completely suppress viral replication in the absence of ART [128]) have higher levels of both CD8^+^ and CD4^+^ T-cells producing IL-2 and IFN-γ [128]. Conversely, ECs had lower levels in both the breadth and magnitude of their HIV-specific CD8^+^ T-cell responses [128], similar to the highly exposed but non-infected sex workers in Kenya [125].

Despite evidence that CD8^+^ T-cells control virus replication, studies have also challenged this assertion. First, no correlation was found between the levels of IL-2- and/or IFN-γ-producing CD8^+^ T-cells and protection from disease progression [124]. However, these samples were collected after the VL set point, when CD8^+^ T-cells could also have played a role in controlling acute viral replication. Second, the in vivo depletion of CD8^+^ T-cells during the chronic SIVsmm infection of SMs had no significant impact on the control of viral replication [129]; other studies also found no correlation between cellular responses and VLs [130,131]. Yet, CD8^+^ cell depletions during the acute infection of natural hosts identified a potential role of cell-mediated immunity in the control of acute viral replication during natural SIV infections [121,132].

#### 2.2.2. CD4^+^ T-Cells

CD4^+^ T-cells become activated when they bind to antigens presented by MHC class II molecules and secrete cytokines [110], such as IL-2, that help boost the function of the CD8^+^ T-cell response against intracellular viral pathogens and also aid the humoral immune response [110]. During persistent viral infections, such as HIV, there is a lack of a CD4^+^ T-helper response against the virus, which may contribute to CD8^+^ T-cell response dysregulation. Similarly, specific CD4^+^ T-cell responses directed against HIV are low [133,134]. Meanwhile, studies failed to establish a clear correlation between CD4^+^ T-cell responses against HIV and protection from AIDS, probably as a consequence of assessing the immune responses during chronic infection [135].

Based on observations that HIV-specific CD4^+^ T-cell responses increase in PWHs receiving ART [136] and ECs [137], it is argued that specific CD4^+^ T-cell responses initiated early in infection may be beneficial for disease outcome [135]. Furthermore, in PWHs receiving ART, as well as in ECs, an increase in effector CD4^+^ T-cells (such as IL-2^+^ CD4^+^ and IL-2^+^ IFNγ^+^ CD4^+^ T-cells) has been observed to be correlated to lower VLs [138]. However, in spite of the increased levels of HIV-specific CD4^+^ T-cell responses in PWHs undergoing ART, residual levels of viral replication indicate that the CD4^+^ T-cell response against HIV is not sufficient to control pathogenesis [134]. Conversely, in vivo CD4^+^ T-cell depletion leads to increased VLs, indicating that CD4^+^ T-cells could be more important as a source of viral replication rather than as a factor controlling viral replication [139,140,141].

Regardless of their exact role in disease progression, CD4^+^ T-cell depletion is a hallmark of HIV infection. In PWHs and NHPs that progress to AIDS, effector memory CD4^+^ T-cells are preferentially depleted [135], with a massive depletion of activated memory CD4^+^ T-cells in the lamina propria of the gut [18,32,34,35,36] and, to a lesser extent, in the peripheral blood and lymph nodes (LNs) [35,36,142]. CCR5^+^ CD4^+^ T-cells (target cells) are also preferentially depleted from the gut [143,144], as expected, given the higher levels of CCR5^+^ CD4^+^ T-cells at the mucosal sites. Note that the original SHIVs were mainly CXCR4 tropic, and preferentially depleted CD4^+^ T-cells from the LNs [18]. ART does not completely restore mucosal CD4^+^ T-cells [144]. However, the acute depletion in the gut is not predictive of disease outcome [50,56].

Particularly important in the gut mucosa are the effector Th17 CD4^+^ T-cells, which help control mucosal inflammation and maintain mucosal health and integrity [111]. They are also preferentially depleted during HIV and SIV infection [145,146,147,148].

### 2.3. Innate Immunity during HIV Infection

Innate immunity is the first responder to microbial pathogens. Innate responses are mounted within hours after infection to limit the spread of the pathogen. These non-specific responses help slow the advance of an infection so that the adaptive immune response has time to respond to the threat [149,150]. At least four types of immune cells are involved in the innate immune response: dendritic cells (DCs), γδ T-cells, NKs, and monocytes. These cells are important as they function as early effector cells, but they are also important later in infection, when they regulate the adaptive immune response [149]. The innate cellular effectors recognize invading organisms via pattern recognition receptors, allowing them to distinguish microorganisms and damaged cells from healthy cells [151]. Meanwhile, anti-HIV antibodies, and sometimes HIV particles, initiate the “classical pathway” of complement activation [152,153]. The complement system does not mediate protection against HIV-infected cells [88], although it may be critical during acute viral replication [87].

### 2.4. T-Cell Immune Activation and Inflammation

The immune system can control HIV infection, as illustrated by ECs [154], and one would expect that increased activation of the immune system is beneficial to fighting an infection. However, chronic activation of the immune system is not beneficial to the host, but rather is the root cause of the deficient immune responses that eventually lead to AIDS [155]. In PWHs, increased levels of immune activation in both CD4^+^ and CD8^+^ T-cells correlate with increased disease progression rates [156,157]. During the early phase of HIV infection, the levels of CD8^+^ T-cell activation are normalized through ART [158]. B-cells are also activated and dysregulated during HIV/SIV infection [159,160]. Likewise, pathogenic SIV infections also display increased levels of immune activation, and these upregulated levels are highly predictive of disease progression [18,37,41].

The central roles of T-cell immune activation and inflammation as the main factors behind HIV/SIV disease progression are supported by the observation that both of these parameters are kept at bay in natural hosts of SIV infection [52,54,55,56,58,67,161], which hinges on the ability of the natural hosts of SIVs to maintain a healthy mucosal barrier throughout the SIV infection [67,68] and prevent microbial translocation [10]. Damage to the mucosal barrier or increases in systemic inflammation have the potential to alter the course of SIV infection in natural hosts [161,162].

### 2.5. Regulatory T-Cells

Tregs downregulate immune responses and are defined by the expression of CD25^+^ and FoxP3^+^ markers from CD4^+^ T-cells [112]. Other markers expressed on the surface of Tregs include GITR and CTLA-4 [112]. The immune response from CD4^+^ and CD8^+^ T-cells is increased when Tregs are depleted from PBMC cultures [163]. Also, Tregs are increased in the LNs, tonsils, and intestines of untreated PWHs, but decrease with the initiation of ART, indicating that Tregs may benefit virus persistence more so than they benefit the host [164,165,166]. Tregs also aid in the formation of collagen deposits in lymphatic tissues, causing further damage to the host [167]. In infant SIVmac251-infected RMs, there is a large number of Tregs that may suppress SIV-specific CD4^+^ T-cell responses [168]. Considering all of this evidence, it is unclear whether Tregs may also be damaging to the host by downregulating specific anti-HIV responses [169]. A loss of Treg function has been reported in PWHs that progressed to AIDS [170,171]. Other studies have found that the loss of Tregs results in further viral persistence and increasing levels of immune activation [172,173,174]. A decrease in TGF-β1 and IL-2 in the LNs was observed in SIV-infected CyMs, suggesting a decrease in Tregs [175]. The levels of FoxP3 increase during the SIVagm infection of AGMs, which is correlated with lower levels of immune activation in natural hosts during SIV infection [52].

While many Tregs are developed and differentiated in the thymus (tTregs), CD4^+^ T-cells can express FoxP3 de novo when naïve T-cells are stimulated, most notably with TGF-β1; in vitro, these novel Tregs are referred to as induced Tregs (iTregs), while in vivo they are referred to as peripheral Tregs (pTregs). Initially, it was thought that iTregs were suitable surrogates for pTregs. iTregs likely do not represent the true state of pTregs in vivo, as full FoxP3 TSDR methylation does not occur in TGF-β-induced Tregs, which are thus unstable, with little suppressive function [169,176]. pDCs previously exposed to HIV can convert non-Tregs into functional pTregs, while unexposed pDCs do not. Similarly, DCs resident in the LNs can induced non-Tregs to become pTregs in untreated PWHs, while this is not seen following ART. In RMs, mDCs from chronically SIV-infected animals more readily induced conversion to pTregs compared to those from non-infected RMs [177].

Altogether, there is extensive evidence to support the idea that T-cell immune activation and inflammation are the driving forces behind HIV’s progression to AIDS [155]. Immune activation has several mechanisms, including (i) the activation of the innate immune system by the virus and subsequently through innate immunity; (ii) the direct activation of T-cells via viral peptides; (iii) microbial translocation from the intestine to the peripheral blood; (iv) other pathogenic coinfections; (v) the induction of non-specific immune activation by inflammatory cytokines; and (vi) the dysfunction of Tregs [155]. However, the functional differences between iTregs in vitro and pTregs in vivo do not permit the assessment of the precise impact of these induced Tregs on HIV/SIV infection [169,176,177].

## 3. Experimental Dissection of Immune Responses in SIV Infection

With the advent of monoclonal antibody (mAb) technology, it became possible to dissect the immune response to SIV by specifically depleting selected cell populations from NHPs in vivo to clearly delineate immune correlates of protection for HIV infection. Several other reagents had already been used previously to this same effect. While largely successful, some limitations relative to these crude previous approaches to study SIV pathogenesis have also been reported. Here, we discuss the results obtained during the immune cell depletion studies performed thus far and various perspectives on how to improve the current technology to more effectively target specific cell populations (Table 1).

### 3.1. CD8^+^ Cell Depletion In Vivo

CTL responses are critical to control HIV and SIV replication, particularly during acute infection [114,116,117,118,178,179,180,181,182,183,184,185,186,187,188,189,190,191,192,193,194,195,196,197]. There is, however, no clear inverse correlation between SIV viremia and SIV-specific cell-mediated immune (CMI) responses [121,196].

In vivo CD8^+^ cell depletion studies have been carried out using a variety of monoclonal antibodies (mAbs): OKT8F, cM-T807, or TRX2.

OKT8F is a murine monoclonal antibody that depletes CD8^+^ T-cells [115,198,199] through complement-mediated cytolysis [115]. OKT8F requires a smaller dosage than cM-T807 to mediate similar effects in terms of the duration and magnitude of CD8^+^ T-cell depletion. Briefly, only 2 mg/kg/day of OKT8F given for three consecutive days can deplete 99% of circulating CD8^+^ T-cells for 8–10 days [115,118,198,200,201,202,203].

A limitation to the use of OKT8F is that, due to the murine portion of the OKT8F Fc region [198], it is not specific and may deplete up to 51% of CD4^+^ T-cells in SIV-infected RMs [115,198]. The use of OKT8F for CD8^+^ cell depletion in SMs can also result in a transient CD4^+^ T-cell depletion. This has occurred even in non-SIV infected SMs, suggesting that the increased viral replication resulting from the depletion of the CD8^+^ T-cells was not responsible for the observed CD4^+^ T-cell depletion [115,129].

In contrast to OKT8F, which is a strictly murine antibody, cM-T807 is a chimeric mouse–human mAb specific for the CD8 marker present on CD8^+^ T- and NK cells [182,204]. The variable regions of the heavy and light chains of cM-T807 are derived from the murine M-T897 hybridoma, which are ligated to the human genes for the γ1 heavy chain and κ light chain [182]. About a decade ago, the Nonhuman Primate Reagent Resource “rhesusized” this mAb by replacing the human γ1 heavy chain and κ light chain with their RM counterparts (M-T807R1). M-T807R1-induced CD8^+^ T-cell depletion likely occurs through Fc-mediated interactions between the antibody and macrophages or granulocytes, or by complement-mediated cytolysis [182]. The duration of depletion by cM-T807 in circulation depends on several factors: (a) the number of treatments; (b) the age of the animal, with longer CD8^+^ T-cell depletion being achieved at similar doses in RMs older than 5 years in comparison with RMs aged 2 years or younger [182]; and (c) the stage of infection, with depletions being 17 to 21 days long during the acute infection stage versus 9–14 days long during the chronic infection stage [118]. CD8^+^ T-cell depletion induced by MT807R1 is more robust, more pronounced, and slightly longer than that obtained with cM-T807 [205,206,207]. Differently from OKT8F, cM-T807 and MT807R1 do not significantly impact CD4^+^ T-cells and B-cells [118,182,204].

T87PT3F9, an anti-human mouse monoclonal antibody [208], was administered at a concentration of 2 mg/kg twice at an interval of 7 days. This mAb was only partially efficient, depleting less than 75% of the circulating CD8^+^ T-cells and only about 50% of the CD8^+^ T-cells from the LNs for two weeks [208]. This antibody is not thought to significantly impact CD4^+^ T-cells [208]. No data are yet available regarding its effectiveness in eliminating CD8^+^ T-cells in the intestine. Compared to the other antibodies, namely OKT8F, cM-T807, MT807R1, and TRX2 (see below), T87PT3F9 is the least effective in eliminating CD8^+^ T-cells.

Finally, TRX2 is a humanized anti-human mAb specific for CD8^+^ T-cells [209]. Only one study has used TRX2 for CD8^+^ T-cell depletion in SIV-infected NHPs; the only other available study used this mAb to avoid kidney transplant rejection in baboons [210]. Three TRX2 treatments of 3 mg/kg on days 0, 1, and 4 depleted >99.9% of circulating CD8^+^ T-cells for approximately two weeks, while eight TRX2 treatments of 3 mg/kg on days 0, 1, and 3; 6 mg/kg on days 6, 10, and 13; and 9 mg/kg on days 16 and 20 depleted >99.7% of the CD8^+^ T-cells over a 7 week period [209]. Six treatments of TRX2 on days 0, 1, 3, 6, 10, 13, and 17 showed significant levels of depletion for approximately five weeks, with the exception of one animal for whom the treatments only depleted 88.7% of the CD8^+^ T-cells [209]. Notably, TRX2 did not cause significant changes in the CD4^+^ T-cells [209]. Overall, both TRX2 and MT807R1 are comparable in their ability to deplete circulating CD8^+^ T-cells, with the only difference being that TRX2 has to be administered over an extended period of time.

In tissues, the effectiveness of CD8^+^ cell depletion is more limited than in circulation. Neither TRX2 (90–100% efficacy) nor MT807R1 (80–100% efficacy) can completely deplete CD8^+^ cells from the LNs [118,179,182,202,209]. The efficacy of CD8^+^ cell depletion in the LNs improves with the addition of more rounds of TRX2 administration [209]. Even though depletion is not complete in the LNs, its duration is equivalent to or longer than that in the periphery [118,209]. Furthermore, neither cM-T807 nor MT807R1 effectively deplete CD8^+^ cells in the gut: in fact, only 37% of the CD8^+^ T-cells were depleted in rectal biopsies in cM-T807-treated RMs [200]. Increasing the dose of cM-T807 to 50 mg/kg and multiple administrations did not improve intestinal CD8^+^ cell depletion [202]. However, specific CD8^+^ responses to Gag were delayed in the intestine, even though depletion was incomplete [202]. A disadvantage of using cM-T807 is the uneven distribution of CD8^+^ cell depletion in vivo, with certain tissues experiencing a greater amount of depletion than others [200]. OKT8F induced significant depletions of CD8^+^ cells of the bone marrow in both RMs and SMs [129,211]. While the maximum CD8^+^ T-cell depletion in the LNs of SMs was 80%, depletion was more effective in infected SMs versus non-infected SMs [129]. None of these studies monitored the effectiveness of OKT8F in depleting cells in the intestine.

#### 3.1.1. CD8^+^ Cell Depletions during Acute SIV Infection

The first CD8^+^ cell depletion studies were performed in SHIV-infected RMs, and consisted of the administration of the depleting antibody at 7 and 14 dpi or 3 and 4 dpi using T87PT3F9. Even though this antibody was not as effective in depleting CD8^+^ cells compared to other mAbs, the authors reported an increase in plasma levels of the p27 Gag antigen in NHPs receiving the anti-CD8 antibody, but not in controls, and an increase in the DNA copies of the CD4^+^ T-cells in circulation (10^3^–10^5^ SIV DNA copies/10^6^ cells versus 10^3^ SIV DNA copies/10^6^ cells in controls) and LNs (10^4^–10^6^ SIV DNA copies/10^6^ cells versus 10^3^ SIV DNA copies/10^6^ cells in controls) [208].

CD8^+^ T-cell depletion with cM-T807 in acutely SIVmac251-infected RMs [118,202] resulted in two depletion patterns, with half of the animals showing complete depletion for less than 21 days (short-term depletion), and the other half showing a more prolonged (28–35 days) CD8^+^ cell depletion (long-term depletion) [118,202]. No difference was observed in the peak VL levels between the CD8-depleted RMs and controls (10^7^ to 10^8^ RNA copies/mL) [118]. However, the post-peak viremia had very different dynamics: the CD8^+^-cell-depleted RMs maintained high levels of viral replication longer than the controls, which reached set-point VLs of 10^5^ to 10^7^ at 21 to 28 dpi. Meanwhile, the short-term CD8^+^-cell-depleted RMs reached these levels at 28–35 dpi, while the long-term-depleted RMs never reached a set point [118,202]. Furthermore, a delay of less than two weeks for the appearance of SIVmac-specific CTLs was observed in RMs with short-term depletion compared to controls, while a longer delay, or no appearance of SIV-specific CTLs, was observed for RMs with the long-term CD8^+^ cell depletion in the periphery, LNs, and intestine [118,202].

Note that during acute SIV infection, the appearance of SIV-specific CTLs was correlated with virus decline in RMs with the Mamu-A*01 MHC class I allele that were not subjected to CD8^+^ T-cell depletion [118,212]. Meanwhile, the CD8^+^-cell-depleted RMs exhibited a greater loss of memory CD4^+^ CD45^neg^ CCR5^+^ cells in both the intestine and LNs compared to controls, with no recovery of these cell types [202]. The RMs depleted of CD8^+^ T-cells progressed to AIDS more rapidly than the controls (average survivals of 114, 240, and >297 dpi for long-term-depleted RMs, short-term-depleted RMs, and controls, respectively) [118]. These results demonstrate the importance of CD8^+^ T-cells for the initial control of viremia during acute SIV infection [213].

In AGMs, cM-T807 administration during the acute infection stage (mAb administration at 0, 6, and 13 dpi) resulted in an increased peak of viral replication and a delayed control of viral replication up to 42 dpi. However, in contrast to macaques, the SIVagm-infected AGMs that were depleted of CD8^+^ T-cells during the acute infection stage were able to control viral replication after the recovery of CD8^+^ T-cells and avoid disease progression to AIDS [132].

#### 3.1.2. CD8^+^ Cell Depletion during Chronic SIV Infection

Three daily administrations of OKT8F (2 mg/kg) to chronically SIVmac-infected RMs successfully depleted up to 99.9% of their circulating CD8^+^ T-cells for up to 10 days [115]. CD8^+^ cell depletion resulted in a VL spike to 10^5^–10^8^ SIVmac RNA copies/mL [115]. With the rebound of CD8^+^ T-cells, the VLs returned to pre-treatment levels (10^3^–10^5^ vRNA copies/mL) in most of the OKT8F-treated RMs [115], suggesting that while CD8^+^ T-cells may be important for controlling chronic SIVmac replication, other components of the immune system may also contribute or assist the CTL response to control viral replication.

Better results have been observed when CD8^+^ T-cell depletion was performed with cM-T807 in SIVmac-infected RMs at 270 dpi: the depletion lasted longer than with OKT8F, yet it was less effective compared to the depletion performed during acute SIVmac infection [118]. Nevertheless, VL spikes of 1–4 logs (to 10^6^–10^7^ RNA copies/mL) were observed [118]. As the CD8^+^ T-cells rebounded to pre-depletion levels, the VLs decreased to the pre-depletion set points (10^3^–10^6^ SIV RNA copies/mL) in the majority of animals. In several cases (occurring after both OKT8F and cM-T807), a second VL spike occurred after the rebound of the CD8^+^ cell population [115,118]. Contrary to acute CD8^+^ T-cell depletion studies, depletions performed during chronic SIV infection did not seem to increase progression to AIDS, indicating that other mechanisms are important for controlling disease progression during this stage of infection.

#### 3.1.3. CD8^+^ Cell Depletions during Vaccine Studies

As CD8^+^ T-cells have been deemed necessary for the control of virus replication based on the CD8^+^ cell depletion experiments described above, numerous vaccine studies have focused on eliciting CTL responses. CD8^+^ cell depletion studies were performed concurrently during vaccine studies to assess the role of CD8^+^ cells in mediating protection. The many vaccine trials involving various vaccine formulations displayed different levels of protection upon challenges with SIVmac, which can be grouped into several categories.

First, RMs immunized with SIVmac239Δnef and challenged with SIVmac251 showed protection from SIVmac251 based on repeated negative PCR tests over a two-year period [198]. Then, these RMs were subjected to OKT8F-induced CD8^+^ T-cell depletion and SIVmac239Δnef loads increased by 2–3 logs in plasma and by 0.6–3 logs in the LNs [198]. With the rebound of CD8^+^ cells, the SIV mac239Δnef VLs returned to pre-depletion levels [198]. CD8^+^ T-cell restoration was also associated with increased SIV-specific CD8^+^ IFN-γ responses in a fraction of RMs [199]. Interestingly, in spite of the fact that virus control was lost and the vaccine virus showed robust replication in this setting, no wild-type SIVmac251 (the challenge virus) was detected throughout the intervention. As such, CD8^+^ cell depletion successfully demonstrated that the SIVmac239Δnef immunization protected the RMs from the challenge with SIVmac251 and that CD8^+^ T-cells were important in controlling SIVmac239Δnef replication [198]. Conversely, when RMs vaccinated with the live attenuated strain SIVmac239Δ3 received cM-T807, CD8^+^ T-cell depletion did not cause an increase in SIVmac239Δ3 replication [198]. The lack of replication of SIVmac239Δ3 was attributed to the highly attenuated nature of this particular strain [214].

In another SHIV study, RMs were primed with DNA containing SHIV 89.6 proteins as well as DNA primed with GM-CSF and a modified vaccinia Ankara boost (MVA) expressing SHIV 89.6 proteins followed by a challenge with SHIV89.6 [203]. While these animals showed protection from SHIV89.6 infection [203], CD8^+^ cell depletion performed 2 years post SHIV challenge resulted in the VLs increasing from undetectable levels to 10^4^–10^6^ RNA copies/mL [203]. Interestingly, following depletion, the increases in CD8^+^ IL-2^+^ IFN-γ^+^ cells and the levels of memory CD8^+^ cells were higher than those observed post challenge [203]. Similarly, RMs vaccinated with DNA primed with HIV Gp160, challenged with nonpathogenic SHIV-vpu^+^, and then challenged again with SHIV89.6 [215] showed a rebound of VLs upon CD8^+^ cell depletion [215].

In a third pathogenic scenario, RMs received an SIV vaccine based on rhesus cytomegalovirus (RhCMV) vectors, which established indefinitely persistent, high-frequency, SIV-specific effector memory T-cell (TEM) responses at potential sites of SIV entry and replication in the RMs. When challenged with the highly pathogenic SIVmac239, the RMs became infected, but then stringently controlled the challenge virus [216]. In these animals, CD8^+^ cell depletion with cM-T807 did not result in any rebound of the controlled challenge virus [216].

Meanwhile, CD8^+^ cell depletion was also employed to monitor the CTLs’ efficacy in protecting against superinfection in RMs infected with the attenuated strain SIVmacC8. The TRX2 depleting antibody was used to this end [209]. Upon CD8^+^ cell depletion, the VLs increased by 1–2 logs, peaking at 10^6^–10^7^ viral RNA copies/mL, higher than in controls (10^5^–10^6^ viral RNA copies/mL) [209]. The depleted RMs were then challenged with SIVmacJ5 and three-quarters of them were protected from super-infection, suggesting that CTLs do not play a role in protection from super-infection [209].

Finally, RMs vaccinated with adenovirus–SIV recombinants and boosted with peptides consisting of the SIV envelope corresponding to the CD4-binding region on SIV were challenged with SIVmac251 [200]. Then, those which were protected were re-challenged with SIVmac251 one year after the first challenge [200] and then subjected to cM-T807-mediated CD8^+^ cell depletion [200]. This resulted in an increase in VLs in both the peripheral blood and intestines that was resolved by the rebound of CD8^+^ T-cells. However, in the LNs, the high levels of virus replication were not decreased through the restoration of SIV-specific CD8^+^ T-cells [200].

Other vaccine studies have assessed the role of CD8^+^ cells in protection by performing depletions prior to the SIV challenges. This approach resulted in a sustained increase in acute viral replication compared to controls (average VLs of 10^6^–10^7^ SIV RNA copies/mL versus 10^5^ SIV RNA copies/mL) [214]. Meanwhile, in a study using an SIV DNA vaccine and boosting with MVA-containing SIV genes, CD8^+^ cell depletion prior to a challenge with SIVmac251 resulted in peak VLs similar to those observed in controls that were mock-vaccinated (10^8^–10^9^ SIV RNA copies/mL) and progressed to disease more rapidly [217], indicating the vital nature of CD8^+^ cells in mediating protection from SIV during the acute infection.

The protective role of CD8^+^ T-cells in the genital tract of RMs vaccinated with live attenuated SHIV89.6 was evaluated by depleting CD8^+^ T-cells with 50 mg/kg of cM-T807, followed by a vaginal challenge with SIVmac239 [218] on the same day. In these RMs, the VLs at day 14 post challenge were similar in the CD8^+^-cell-depleted group to those in the non-vaccinated controls and higher than those in the immunized non-depleted group [218].

#### 3.1.4. Dynamics of Cell-Mediated Immune Responses during CD8^+^ Cell Depletions

SIV-specific Gag CD8^+^ T-cells increased after both acute and chronic CD8^+^ cell depletions [118]. However, in vaccine studies, this increase did not correlate with protection (or lack thereof) from SIV challenges [199,209]. Interestingly, a moderately protected RM did not display any SIV-specific Gag CD8^+^ T-cell activity, but did show a modest increase in antibodies specific for SIV Gag Gp130 and P27 [199]. Additional research also failed to find a correlation between CTL precursors and protection from the attenuated SIVmac239Δnef challenge [219], showing that even though CD8^+^ T-cells have been associated with increases in VLs during pathogenesis or vaccine studies, they are probably not the only correlate of immune protection that can prevent disease progression.

#### 3.1.5. Dynamics of Humoral Immune Responses during CD8^+^ Cell Depletions

Experimental CD8^+^ cell depletions also permitted the assessment of the contribution of antibodies to SIV control. The levels of binding antibodies increased during the CD8^+^ cell depletions. CD8^+^ T-cell depletion after immunization with SIVmac239Δnef and a subsequent challenge with SIVmac251 resulted in moderate to large increases in anti-Gp130 and anti-P27 antibody titers two weeks after depletion in the majority of CD8-depleted RMs [199] and a rise in NAbs [199]. However, when RMs were additionally challenged with SIVmac055, those RMs that were protected showed a higher baseline of antibodies and a subsequent increase in the titers of specific antibodies for SIV Gp130 and P27 [199]. An increase in anti-Gp130 and anti-P27 antibody titers was observed 14 days after a challenge with SIVmacJ5, although the anti-P27 antibody titers were not different from those of controls [209].

However, the dynamics of Nabs are somewhat different from those observed for binding antibodies. In CD8^+^ cell depletion studies performed during acute SIVmac infection, no Nabs were detected during depletion (as the Nab responses mature slowly during infection); conversely, NAbs increased when CD8^+^ depletion was performed during chronic SIVmac infection [118]. Meanwhile, NAbs did not confer protection to previously immunized animals challenged with SIVmac055 or SIVmac251 [199,200]. However, NAb levels were above baseline at 56 days after a challenge with SIVmacJ5 initiated during CD8^+^ T-cell depletion [209]. A negative correlation was found between NAb levels and the peak of virus replication during CD8^+^ cell depletion in SIVmac251-challenged RMs, suggesting that NAbs may play some role in the control of virus replication, albeit a less active role than that of CD8^+^ T-cells [214].

Altogether, these results suggest a secondary role, if any, of NAbs in relation to the CTL response in controlling viremia post CD8^+^ cell depletion. It is possible that they function in a different role entirely in SIV replication by protecting CD4^+^ T-cells that are targeted by the virus [203]. These results indicate that NAbs are potentially not as important in the control of chronic viral infection.

#### 3.1.6. Impact of CD8^+^ Cell Depletion on Other Immune Cell Populations

A certain degree of non-specific cell depletion was observed during treatments with mAbs designed to deplete CD8^+^ T-cells. This non-specificity was dependent upon the product used, with OKT8F being the least-specific antibody. Thus, OKT8F could induce a certain degree of CD4^+^ T-cell depletion [115]. Significantly less impact on CD4^+^ T-cells was observed when using other mAbs.

One of the major limitations of CD8 depletion studies is that they rely on the nature of the CD8^+^ molecule. Most of the studies carried out employed CD8^+^-depleting mAbs recognizing the CD8α chain. Thus, these antibodies did not target CD8^+^ T-cells exclusively, as NKs also express the CD8α chain and were also depleted [129,203,204,218]. NK depletion may have also contributed to the increase in viremia in the studies above. The idea that NKs could play a potential role in protection in HIV infections has been postulated after the observation that some intravenous drug users have greater levels of NK cytolytic activity and do not become infected after exposure to HIV [220]. Also, NKs were involved as one of the mechanisms of the HIV-post-treatment control in the Visconti trial (a protocol that involved aggressive ART administered early in the infection for over 2 years, which resulted in no viral rebound upon cessation of ART in 15% of cases) [221] and in viral control after acute HIV infection [222,223]. However, RMs challenged with SHIV and depleted of CD8^+^ T-cells post SIV challenge exhibited no detectable levels of circulating NK cells when the VLs were declining [203].

More recently, a rhesus IgG1 recombinant anti-CD8β mAb [CD8β255R1] was engineered and produced by the NIH Nonhuman Primate Reagent Resource. This antibody was employed to assess the contribution of CD8αβ^+^ T-cells in controlling live attenuated SIV replication during chronic infection and subsequent protection from pathogenic SIV challenges [224]. Unlike the studies in which a CD8α-specific depleting mAb was employed, CD8β255R1 selectively depleted CD8αβ^+^ T-cells without depleting NK cells and γδ T-cells that also express CD8α [224]. Following infusion with CD8β255R1, plasma VLs transiently increased coincidentally with declining peripheral CD8αβ^+^ T-cells. Yet, virus suppression to pre-depletion levels occurred even prior to the rebound of peripheral CD8αβ^+^ T-cells [224]. The authors interpreted these results as a reflection of the limited ability of CD8β255R1 to deplete CD8αβ^+^ T-cells in the LNs [224]. Interestingly, CD8αβ^+^ T-cell depletion with CD8β255R1 did not eliminate the protection afforded by live attenuated SIV vaccination [224].

Finally, it was reported that increases in viremia in SIV-infected RMs after anti-CD8 treatment may be partially related to an increased activation of CD4^+^ T-cells (specifically those residing in lymphoid or mucosal tissues, but not in the blood compartment sampled in these studies). The increase in CD4^+^ T-cell immune activation may occur through various mechanisms, such as the administration of large doses of a foreign protein (i.e., the depleting mAb), the inflammatory signal resulting from the loss of large numbers of CD8^+^ T-cells (in lymphoid tissues), the reactivation of latent viruses (i.e., CMV) resulting from the loss of CD8^+^ T-cells, or increased CD4^+^ T-cell proliferation resulting from a blind attempt to restore the overall T-cell homeostasis [26]. Such activated CD4^+^ T-cells would serve as potential targets for SIV infection.

Thus, while these CD8 depletion experiments suggest a major role of CD8^+^ T-cells in determining the level of virus replication, the contribution of other factors cannot be ruled out by these depletion studies [26]. This question was addressed in a series of experiments consisting of the administration of IL-15 in addition to an anti-CD8 antibody [225]. Through this approach, CD8 depletion was decoupled from excessive immune activation [225]. These studies reported significant increases in VLs, similar to those reported previously in other CD8^+^ cell depletion experiments. It was therefore concluded that the rebound in viral replication after CD8^+^ cell depletion is due to the ablation of CTLs and not to excessive immune activation [225].

#### 3.1.7. CD8^+^ Cell Depletion Studies in Association with ART

While the dynamics of HIV/SIV-specific immune responses, as well as CD8^+^ cell depletion studies, point to the role of CD8^+^ T-cells in controlling HIV, characterizing the contribution of CD8^+^ T-cells on the control of the virus suppressed with ART has the potential to provide an important rationale for exploring approaches to reduce the viral reservoir in PWHs.

Several studies have investigated the role of CD8^+^ T-cells in ART-suppressed SIV infection. SIVmac251-infected RMs were treated with tenofovir (a reverse-transcriptase inhibitor) and then cM-T807 two weeks post infection (pi) [226], three weeks pi, and one week after the beginning of ART. The VLs of the CD8-cell-depleted and tenofovir-treated RMs were significantly higher than those observed in the control group receiving tenofovir only. In the tenofovir-treated, CD8-cell-depleted RMs, VLs were similar to those in the controls that received no ART. However, once CD8^+^ cells began to increase at four weeks pi, two weeks after treatment initiation, the VLs began to decrease at a rate similar to that observed in the tenofovir-treated RMs [226]. Furthermore, RMs that were administered tenofovir 24 h pi with SIVEM660 and re-challenged with SIVEM660 10 weeks and 65–75 weeks pi exhibited on both occasions either complete protection or only a brief replication of the challenge virus [179]. Two of these RMs were then depleted of CD8^+^ cells, which resulted in a transient increase in SIV replication in one animal to approximately 10^3^ vRNA copies/mL and no detectable increase in the plasma viremia for the other [179]. However, the RM for which no plasma viremia could be detected tested positive for the virus in a cell culture supernatant cultured from LN cells [179]. In a separate experiment, additional RMs treated early with tenofovir and challenged with SIVEM660 were challenged later with SIVmac239. The animals had post-peak SIVmac239 VLs of 10^2^–10^4^ RNA copies/mL, which increased by 3–6 logs with the administration of cM-T807 [179]. The VLs fell to near pre-treatment levels with the return of the CD8^+^ cells [179]. Both of these studies show that virus control during ART also depends on a CD8^+^ T-cell response.

CD8^+^ depletions were also performed in RMs receiving ART to investigate the mechanisms through which CD8^+^ T-cells control virus replication. The in vivo effect of CD8^+^ lymphocyte depletion on the lifespan of productively infected cells during chronic SIVmac239 infection was assessed in RMs that were either CD8^+^-lymphocyte-depleted or not, and then treated with ART [227]. Neither early (57 dpi) nor late (177 dpi) CD8^+^ cell depletion before treatment resulted in a measurable increase in the lifespan of either short- or long-lived productively infected cells in vivo. The study concluded that the presence of CD8^+^ lymphocytes does not result in a noticeably shorter lifespan of productively SIV-infected cells, and therefore that direct cell killing is unlikely to be the main mechanism underlying the antiviral effect of CD8^+^ T-cells in SIV-infected RMs with high virus replication [227]. The contribution of CD8^+^ T-cells to maintaining viral suppression under ART was further investigated in SIV-infected RMs treated with short-term (i.e., 8–32 weeks) ART. In these RMs, CD8^+^ T-cell depletion resulted in increased plasma VLs and the repopulation of CD8^+^ T-cells was associated with the prompt reestablishment of virus control. SIV-DNA-positive cells remained unchanged after CD8^+^ cell depletion and reconstitution, yet the frequency of SIV-infected CD4^+^ T-cells before depletion was positively correlated with both the peak and area under the curve of viremia after depletion. Altogether, these results demonstrate a role for CD8^+^ T-cells in controlling viral production during ART [228].

Based on data from these CD8^+^ cell depletion experiments, a viral dynamic model to fit the VLs was created. The dynamics of the latent reservoir and the SIV-specific effector cell population, including their exhaustion and their potential cytolytic and noncytolytic functions, were evaluated and the model showed that the latent reservoir significantly contributes to the size of the peak VL after CD8^+^ cell depletion, while drug efficacy plays a lesser role [228,229]. The model also suggested that the overall CD8^+^ T-lymphocyte cytolytic killing rate changes dynamically depending on the levels of antigen-induced effector cell activation and exhaustion. The study concluded that before or without ART, the overall CD8^+^ T-cell cytolytic killing rate is small due to exhaustion. However, after ART, the rate increases due to an expansion of SIV-specific CD8^+^ effector T-cells. The cytolytic killing rate can be significantly larger than the cytopathic death rate in some RMs during the second phase of viral decay with ART [229]. Therefore, this modeling provides a new explanation for the previously puzzling findings that CD8^+^ cell depletion performed immediately before ART has no noticeable effect on the slope of first-phase viral decay during ART [227].

In a different set of experiments, CD8^+^ cell depletions were used to address the very important question of whether CD8^+^ T-lymphocytes control HIV/SIV infections through cytopathic or noncytopathic mechanisms [230]. While the studies outlined above highlighted noncytopathic effects, cytopathic effects of CD8^+^ T-cells can occur before viral production. To examine the role of CD8^+^ T-cells prior to virus production, RMs infected with SIVmac251 received either an integrase inhibitor combined with a CD8^+^ cell depletion (MT807R1), or each reagent separately. A mathematical model was created that included infected cells pre and post viral DNA integration to compare different immune effector mechanisms. The model predicted that the CD8^+^-cell-mediated reduction in viral production (noncytolytic) best explains the viral profiles across all RMs, while an effect in killing infected cells pre integration (cytolytic) was supported in some of the best models. As such, this study suggests that CD8^+^ T-cells have both a cytolytic effect on infected cells before viral integration, and a direct, noncytolytic effect by suppressing viral production [230].

Finally, CD8^+^ cell depletions have been combined with ART to investigate the role of CD8^+^ T-cells in the formation of the latent reservoir in SIV-infected RMs [231]. CD8^+^ cell depletion was performed either before infection or before early (i.e., 14 dpi) ART initiation and resulted in a slower decline of viremia under ART, indicating that CD8^+^ lymphocytes reduce the average lifespan of productively infected cells during acute infection and early ART, presumably through SIV-specific CTL activity [231]. However, CD8^+^ cell depletion did not change the frequency of infected CD4^+^ T-cells in the blood or LNs, as measured according to the total cell-associated viral DNA or with an intact provirus DNA assay. In addition, the size of the persistent reservoir remained the same when measuring the kinetics of virus rebound after ART interruption [231]. These data indicate that during early SIV infection, the establishment of the viral reservoir that persists under ART is largely independent of CTL control [231].

#### 3.1.8. Impact of CD8^+^ Cell Depletion on Elite Controlled Infections

The Mamu-A*01, Mamu-B*17, and Mamu-B*08 alleles are MHC class I alleles found in Indian RMs that correlate with better control and survival of pathogenic SIV infections [232,233,234,235]. The Mamu-A*01 allele is associated with CTL responses against one particular epitope during the acute phase [233]. In CD8^+^-T-cell-depleted RMs challenged with SIVmac251, virus rebound was more substantial in Mamu-A*01-positive RMs compared to Mamu-A*01-negative RMs [214]. Moreover, stronger CD8^+^ T-cell SIV-specific responses have been observed in vaccinated Mamu-A*01-positive RMs [203,214]. The administration of a single dose of 50 mg/kg cM-T807 to four EC RMs that controlled SIVmac239 replication to below detectable levels for 1 to 5 years resulted in 2–4 log increases in VLs [204]. After the return of VLs to the set point, PBMCs isolated from these RMs and depleted of either NKs or CD8^+^ T-cells were cultured in vitro and this culture system showed increased viral replication, leading to the conclusion that both CD8^+^ T-cells and NKs are important players in controlling viral replication in ECs [204]. Viral control after depletion was also associated with an increase in subdominant CD8^+^ T-cell responses and an increase in the Gag CD4^+^ T-cell response [204]. Similar results were observed after CD8^+^ cell depletion using M-T807R1 [207]. Furthermore, our CD8^+^ cell depletion studies in SIVagm-infected RMs, in which VLs were completely controlled for >4 years, showed a 3–4 log increase in viral replication for the duration of the CD8^+^ cell depletion [236]. To assess the replication competence of the rebounding SIVagm, plasma collected at the peak of the virus rebound from these CD8^+^-cell-depleted RMs was passaged in naïve RMs. These RMs became infected with a pattern of SIV infection that was indistinguishable from the original infection. Altogether, these studies point to a potential role of cellular immune responses in the control of viral replication in ECs.

CD8^+^ cell depletion studies have also been performed in natural hosts to assess the contribution of this immune cell population in establishing a nonprogressive status. In SMs, either OKT8F or cMT807 were employed to deplete the CD8^+^ T-cells during chronic infection [129]. CD8^+^ cell depletion resulted in a VL increase of 1–2 logs. The authors also found a correlation between increases in VLs and activated CD4^+^ Ki67^+^ T-cells and concluded that these increases were more likely caused by a reactivation of latent viruses, such as CMV, that reemerged due to the lack of CD8^+^ cells, thus leading to an increase in activated CD4^+^ T-cells [129]. In AGMs, CD8^+^ cells were depleted during both acute and chronic infection either alone [132] or in combination with CD20^+^ B-cell depletion [237,238]. Although these interventions generally result in transient increases in VLs, none of them altered the overall course of SIV infections in the natural hosts of SIVs.

### 3.2. B-Cell Depletion Studies

Humoral immune responses are critical for preventing, controlling, and eradicating HIV infection, as demonstrated by the numerous studies employing bNAbs [239,240,241,242,243,244,245,246,247,248,249,250,251,252,253]. The results of the Thai Phase III HIV Type 1 Vaccine Trial (RV144) demonstrated that antibody responses were responsible for the modest protection conferred by the administration of the ALVAC-HIV (vCP1521) prime and AIDSVAX B/E vaccine [8], further fueling the research on antibody responses in HIV infection and leading to the discovery of bNAbs and the design of new strategies for HIV prevention and eradication [254,255].

Animal studies have been instrumental in understanding the efficacy of humoral immune responses for the prevention and control of HIV infection [256]. Intravenously administered antibodies have been shown to protect macaques against intravenous or mucosal SHIV challenges [257,258,259,260]. Topically applied antibodies can also protect macaques against vaginal SHIV challenges [260,261]. Antibody protection is mainly achieved through neutralization (i.e., the antibody’s ability to inhibit viral entry into target cells, thus preventing infection), but it is also achieved through other antiviral effects (i.e., effector functions mediated by the crystallizable fragment of antibody molecules, such as complement activation and ADCC) [88]. Considering their central role in many successful vaccines in the past, antibody-based vaccines were the first choice for vaccine development [7]. However, the resistance of primary HIV isolates to neutralization has been a major hurdle [7], as neutralization activity tends to be rather type-specific and the high sequence variability in Env means that the virus can easily escape [78,262,263]. Nevertheless, a fraction of PWHs develop bNAbs, providing a paradigm to guide vaccine development [254,255,264].

B-cells are key components of the immune system because of their three functions [265]: (i) through antibody production, they provide antimicrobial defenses and mediate tissue inflammation [266]; (ii) through cytokine production, co-stimulation, and antigen presentation, they participate in T-cell activation [266,267]; and (iii) they have regulatory capacities that modulate both cellular and humoral responses. However, the precise role of B-cells in HIV/SIV infection is under debate. During HIV infection, B-cells produce an excess of antibodies and also display increased markers of immune activation [267] and dysfunction [160], as well as increased levels of apoptosis, particularly of memory B-cells [267]. NAbs first appear in low titers after viremia is controlled [200,232]; humoral immune response matures over the following months [89]. A negative correlation can be established between the level of NAbs and plasma VLs [203,214]. Passive immunization studies with serum immunoglobulins administered to chronically SIVmac251-infected RMs that were rapid progressors did not show any change in the VLs after antibody administration, suggesting a limited role for NAbs in the control of viral replication [268]. bNAbs have the potential to control HIV either by preventing the virus from entering the target cells [269,270,271] or by helping the immune system to clear the virus, thus being very promising for cure and remission strategies [249,272,273,274,275].

Given the inverse correlation between NAb titers and disease progression in SIV-infected RMs (i.e., RMs with the highest NAb titers have slower disease progression) [276], B-cell depletions were carried out to directly assess the impact of antibodies on viral replication.

All of these CD20 B-cell depletion studies in NHPs were performed prior and during acute SIV infection. The rationale for this design is that the antibodies are produced by the cells, but the cells themselves do not provide immunity. Since antibodies persist in the circulation for long periods of time, B-cell depletion has a delayed rather than immediate effect on nascent or established antibody titers. Plasma cells also produce antibodies and do not contain a significant amount of CD20 on their surface and they may or may not be depleted with the addition of the B-cell-depleting mAb (Rituximab). This is a slight concern if plasma cells are short-lived, but if they have a relatively long lifespan, this could be a major limitation for CD20 B-cell depletion studies [277]. Therefore, the role of humoral immune responses in controlling SIV replication can only be assessed by depleting B-cells prior to SIV inoculation to prevent antibody development. Only a few such experiments have been reported so far, significantly fewer than the CD8^+^ T-cell depletion studies, and their conclusions have not always been in agreement.

Only one B-cell-depleting agent has been used thus far for SIV pathogenesis studies: a chimeric human–mouse anti-CD20 mAb composed of the constant human IgG1 region, and the variable region of the murine anti-human CD20 B-cell hybridoma, Rituximab (Rituxan, Genentech, San Francisco, CA, USA) [278].

After initial administration of the B-cell-depleting antibodies, B-cells can be continuously depleted to assess the effects of the ablation of humoral immune responses during the chronic phase of infection and also upon disease progression.

There are multiple rationales for specifically targeting CD20 when one attempts to suppress humoral immune responses. CD20 is a hydrophobic membrane-associated phosphoprotein that is highly expressed on the surface of B-cells. It is involved in the early steps of activation and differentiation of the B-cell cycle, and also functions as a possible calcium ion channel [277]. Targeting the CD20 antigen is an ideal approach to deplete B-cells, because this molecule is only expressed on pre-B-cells and mature B-cells [278]. CD20 is not lost from the cell surface during the B-cell maturation process, and there are no detectable levels of CD20 in serum. Finally, CD20 does not internalize when it binds to the anti-CD20 antibody [279].

The mechanisms postulated to drive Rituximab’s effects against CD20^+^ B-cells involve ADCC, complement-dependent cytotoxicity (CDC), and apoptosis stimulation [277,278,279]. In humans, Rituximab is used in the treatment of patients with hematologic disorders: leukemias, lymphomas, Waldenström macroglobulinemia, etc. [280,281,282].

The first CD20^+^ B-cell depletion studies performed in NHPs employing Rituximab were conducted in CyMs during assessments of the drug’s cytotoxicity. The administration of four doses of two different concentrations (0.4 mg/kg and 1.6 mg/kg) resulted in a 98% depletion of peripheral CD20^+^ B-cells for a total of 8 days. Neither dose was successful in completely depleting the B-cells from the LNs; increasing the Rituximab dosage to 16.8 mg/kg weekly for eight weeks resulted in 88–95% depletions of the CD20^+^ B-cells in the LNs, which were maintained for 22 to 36 days after completion of the Rituximab treatment [279].

In RMs, the initial studies used a standard concentration of 20 mg/kg of Rituximab administered over the course of three injections once a week for three weeks. With this approach, circulating CD20^+^ B-cells were depleted for 4 to 8 weeks [283,284,285], an insufficient duration to generate meaningful data; meanwhile, the B-cells rebounded during this treatment. The administration of Rituximab at a higher concentration (50 mg/kg) every three weeks for up to 200 days improved the efficacy of the CD20^+^ B-cell depletion, inducing a persistent depletion of peripheral CD20^+^ B-cells for up to 300 days [286]. The depletion of peripheral CD20^+^ B-cells was complete in all RMs; however, even at this increased dosage, significant tissue CD20^+^ B-cell depletion only occurred in about 50% of RMs [286], and only the complete depletion in tissues was associated with the ablation of antibody production [286]. The efficacy of Rituximab-induced CD20^+^ B-cell depletion can thus be predicted only by their complete tissue depletion [286]. The polymorphisms in the structure of FCγR3A were not predictive for the efficacy of Rituximab-induced CD20^+^ B-cell depletion in RMs, which is different to what has been reported in human patients with certain hematological disorders [280,281,282].

A major limitation of the studies assessing the impact of B-cell responses on viral replication is that most studies used reference strains (SIVmac251 and SIVmac239), which are highly resistant to antibody neutralization. A second limitation of these CD20^+^ B-cell depletion studies is that the significance of their results might be underrepresented. To properly assess the contribution of humoral responses to the control of viral replication, one needs to demonstrate that the antibodies that bind to SIV are, in fact, depleted. The titers of binding anti-SIV antibodies need to be evaluated at the same time as the NAb titers. Also, as noted above, while different surrogate markers of depletion have been proposed, sufficient CD20^+^ B-cell depletion to prevent specific anti-SIV production can only be predicted by the tissue levels of CD20^+^ B-cells. Finally, confounding factors of B-cell depletion should be considered before concluding the impact of humoral immune responses on the dynamics of anti-SIV replication; for instance, a natural depletion of CD20^+^ B-cells via apoptosis may also occur during HIV infection, likely during the acute infection phase, which may also contribute to an undermining of the antibody responses in CD8^+^ T-cell depletion studies [276].

Using SIVmac251, Schmitz et al. found that VL levels were similar in CD20^+^-B-cell-depleted RMs and non-depleted controls [284]. Furthermore, SIV-specific CD8^+^ T-cell activity levels were not different between the CD20^+^-B-cell-depleted and control groups [284]. A negative correlation between NAb titers and plasma VLs was reported in that study [284], indicating a potential, though not major, role of antibodies in controlling viral replication. In the meantime, Rituximab infusion followed by infection with the highly neutralization-sensitive mutant SIVmacM5 derived from the parental strain SIVmac239 had no discernable impact on early viral replication levels between CD20^+^-B-cell-depleted and non-depleted RMs [283]. However, significant differences were observed 30–40 weeks post SIVmacM5 infection, with the majority of CD20^+^-B-cell-depleted RMs showing sustained increased viral replication compared to controls [283]. Significant variability in NAb titers was observed in the B-cell-depleted RMs compared to controls.

In CD20-depleted Mamu-A*01-negative RMs, CD8^+^ T-cells failed to control SHIVDH12R-Clone8 replication, resulting in a sustained loss of CD4^+^ T-cells, high levels of plasma VLs, and disease progression to AIDS [232]. However, while CD4^+^ T-cell depletion also occurred in Mamu-A*01-positive RMs, depletion was not persistent, and the animals did not develop any sign of disease progression [232].

Continuous Rituximab administration to SIVmac239-infected RMs has shown that, while CD20^+^ B-cell depletion did not determine discernable changes in VLs, the duration of disease progression in CD20^+^-B-cell-depleted RMs was significantly shortened compared to non-depleted ones (16 vs. 24 weeks pi, respectively). Similarly, CD20 depletions in SIVsmmFGb-infected PTMs also resulted in a more rapid progression compared to non-depleted controls [287].

Finally, the use of a neutralization-sensitive SIVsmm (D215) strain demonstrated no significant changes in viral replication or disease progression between CD20^+^-depleted or non-depleted RMs [286]. Moreover, a trend for lower VLs was observed in CD20^+^-depleted RMs compared to controls. Cellular immune responses were more pronounced in CD20^+^-depleted RMs than in controls [286].

Prolonged CD20 depletion studies performed in SIVsab-infected AGMs also failed to identify any significant impact of humoral immune responses on controlling SIVsab replication [288].

Altogether, these studies fail to establish a significant contribution of humoral immune responses in controlling SIV replication. When corroborated by passive immunoglobulin infusion results, which also fail to reveal any significant impact of antibodies on controlling viral replication, these results suggest that antibodies play no major role in the control of viral replication. Whether or not the observed trends for the negative impacts of B-cells on the natural history of SIV infection are significant is yet to be determined.

### 3.3. Depletion of Other Immune Cells

#### 3.3.1. Depletion of CD4^+^ T-Cells

In PWHs and macaques, CD4^+^ T-cell depletion is pathognomonic and predictive of HIV/SIV disease progression [289]. T-cell depletion occurs early; it is massive at mucosal sites and is not completely reversed with ART, particularly if ART is initiated late in infection, when T-cell functions are compromised. HIV/SIV target cells are activated by CCR5-expressing memory and effector CD4^+^ T-cells, which are mainly located in the lamina propria of the gut [202]. These cells are rapidly infected and exhausted in the course of the acute HIV/SIV infection. Acute CD4^+^ T-cell depletion occurs mainly due to the massive amount of virus replication, as supported by the observation that CD4^+^ T-cell depletion occurs irrespective of the nature of the HIV/SIV infection: whether it be a progressive, nonprogressive, or controlled infection [56]. Meanwhile, the clinical outcomes of the different types of SIV infections are predicted by the degree of chronic mucosal CD4^+^ T-cell recovery, with no recovery occurring in rapid progressors, and partial, transient recovery occurring in normal and long-term progressors, the degree of which depends on the virus control [32,290].

During the nonprogressive infection of African NHPs that are natural hosts of SIVs, a partial mucosal CD4^+^ T-cell recovery occurs during the chronic infection in spite of the continuous high viral replication [50,56]. In ECs, a complete, albeit very slow, recovery of mucosal CD4^+^ T-cells has also been reported to occur [236]. Early ART does not prevent mucosal CD4^+^ T-cell depletion [291,292], yet it greatly improves their restoration, sometimes to pre-infection levels [292]. Comparative assessments of the dynamics of CD4^+^ T-cells in different models of SIV infection suggest that T-cell immune activation and inflammation, in addition to viral replication, are key drivers of the CD4^+^ T-cell depletion, with immune restoration occurring only when these parameters are minimized [289]. CD4^+^ T-cell depletion is persistent, and recovery is very slow, even when both the virus and T-cell immune activation and inflammation are completely controlled [289]. However, even partial mucosal CD4^+^ T-cell recovery is sufficient for a normal, healthy life in natural hosts [289]. The loss of CD4^+^ T-cell subsets that are protective for the health of the intestinal mucosa also triggers mucosal inflammation and enteropathy, which alter the integrity of the mucosal barrier, leading to microbial translocation, the major driver of T-cell immune activation and inflammation [293]. CD4^+^ T-cell loss is also related to opportunistic infections, cancers, and comorbidities. It is thus critical to preserve CD4^+^ T-cells (through early ART) during HIV/SIV infection. Residual T-cell immune activation and inflammation can even persist in patients receiving early ART, preventing and/or delaying CD4^+^ T-cell restoration.

CD4^+^ T-cell depletion occurs through multiple mechanisms: some are virus-related [289], such as the death of infected cells through the action of HIV/SIV-specific CD8^+^ T-cells [294], cytolysis induced by virus release [295], and programmed cell death [296,297]. As the majority of CD4^+^ T-cells are not HIV/SIV-infected [36], CD4^+^ T-cell depletion is likely driven by other mechanisms. Some of these mechanisms have been extensively investigated: (i) increased apoptosis of non-infected cells exposed to viral antigens [298,299]; (ii) activation-induced cell death [300]; (iii) pyroptosis in cells undergoing abortive infection [301]; and (iv) the destruction of cells captured in neutrophil extracellular traps induced by HIV/SIV infection [302]. CD4^+^ T-cell depletion is thus central to HIV/SIV infection and is not entirely restored in PWHs undergoing ART [147].

As the two drivers of HIV/SIV disease progression, systemic inflammation and T-cell activation and CD4^+^ T-cell depletion are closely interrelated; multiple studies have assessed the impact of various parameters on the progression of untreated HIV/SIV infection, based on the rationale that identifying key driver(s) of disease progression could help establish treatment priorities. However, the exact impact of CD4^+^ T-cell depletion on gut damage and on disease progression can only be assessed by performing experimental CD4^+^ T-cell depletion studies.

Anti-CD4 mAb administration to RMs prior to SIV infection abrogated the post-acute control of viremia, accelerated disease progression [303], and facilitated virus uptake by myeloid cells [141].

One strategy for an HIV cure has been the in vivo depletion of the reservoir through the use of mAbs directed against CD4^+^ T-cells. Yet, CD4R1 mAb successfully depleted 65 to 89% of the circulating CD4^+^ T-cells in RMs and was significantly less effective in depleting the CD4^+^ T-cells from tissues, with only 20 to 50% of the CD4^+^ T-cells from the LNs being depleted [141]. Therefore, considering that only 2% of the CD4^+^ T-cells are in circulation and that depletion in tissues is very limited, it is very unlikely that experimental CD4^+^ T-cell depletion can be a successful strategy for curbing the HIV/SIV reservoir. CD4^+^ T-cell depletion experiments did not impact other cell types such as monocytes and dendritic cells. RMs in which CD4^+^ T-cells were depleted prior to SIV infection showed a similar peak of viremia to that observed in non-depleted macaques, but did not manifest any post-peak decline of virus replication, in spite of the fact that the SIV-specific CD8^+^ T-cell- and B-cell-mediated immune responses were comparable to those observed in controls.

Interestingly, the depleted animals displayed rapid disease progression, which was associated with increased virus replication in non-T-cells, as well as the emergence of CD4-independent SIV envelopes [303]. These results suggest that the antiviral CD4^+^ T-cell responses may contribute to limiting SIV replication. An investigation of the sources of the viral burden and the lifespan of productively infected cells during the CD4^+^ T-cell depletion showed that (i) the set-point VL was two logs higher than that observed in controls; (ii) macrophages took over the function of the CD4^+^ T-cells and constituted 80% of all SIV vRNA^+^ cells in LNs and mucosal tissues in the CD4^+^-cell-depleted RMs; and (iii) proinflammatory monocytes experienced a substantial expansion after the CD4^+^ T-cell depletion [141]. Additionally, CD4^+^ cell depletion resulted in the aberrant activation and infection of microglial cells. Finally, these CD4^+^ cell depletion experiments showed that the lifespan of productively infected cells was significantly longer compared to that observed in controls, but markedly shorter than that previously estimated for macrophages [141].

CD4^+^ T-cells recovered very slowly after depletion in RMs [304]. Recovery was associated with a very high cell proliferation, yet this CD4^+^ T-cell proliferation was not associated with detectable increases in viremia, strongly suggesting that the homeostatic activation of CD4^+^ T-cells is not sufficient to induce virus reactivation from latently infected cells [304]. Meanwhile, the homeostatic reconstitution of the CD4^+^ T-cell pool was not associated with significant changes in the CD4^+^ T-cells harboring SIV DNA, leading to the conclusion that it does not increase the size of the virus reservoir [304].

In another CD4^+^ depletion study, SIV-infected RMs were treated with ART for 93 weeks starting 4 dpi. As ART, the SIV-infected RMs received five to six anti-CD4 antibody administrations and then CD4^+^ T-cells were allowed to recover for 1 year under ART. A slight reduction in the size of the viral burden was reported to occur after the administration of six doses of the CD4-depleting antibody [140]. However, after 1 year, when the ART was stopped, and despite profound CD4^+^ T-cell depletion in the blood and LNs, the time to viral rebound following ART cessation was not significantly delayed in anti-CD4^+^-treated RMs compared with controls [140]. The virus reactivation rates based on rebounding SIVmac239M clonotype proportions were also not significantly different in CD4-depleted RMs [140]. Notably, antibody-mediated depletion was limited in rectal tissue and negligible in lymphoid follicles [140]. These results suggest that, even if robust viral reactivation can be achieved, antibody-mediated viral reservoir depletion may be limited in key tissue sites [140].

CD4^+^ T-cell depletion in chronically SHIV_SF162P3N_-infected Chinese CyMs did not affect viremia over 22 weeks post depletion and resulted in an increased CD4^+^ T-cell proliferation and turnover of macrophages during the early stages of the depletion, followed by a gradual decline to pre-depletion levels. All of this had little impact on the expression of the inflammatory cytokines and CC chemokines associated with disease progression [305].

In conclusion, the net effect of CD4^+^ T-cell depletion in pathogenic SIV infections or macaques was a loss of control of SIV replication and a shift in the virus tropism to macrophages, microglia, and, potentially, other CD4-low cells which all appear to have a shortened in vivo lifespan [141]. Unfortunately, CD4^+^ T-cell depletion with mAbs was not effective enough to significantly curb the viral reservoir.

In natural hosts of SIVs, CD4^+^ T-cell depletion was achieved in chronically SIV-infected SMs with the OKT4A-depleting antibody (10 mg/kg, followed by three additional administrations of 5 mg/kg each on days 3, 7, and 10 after the first treatment) [139]. CD4^+^ cell depletion was significant in the periphery, LNs, and bone marrow. Conversely, the impact of antibody treatment on CD4^+^ cells in mucosal tissues was not as significant. Depletion resulted in lower levels of CD4^+^ T-cells compared to baseline levels, which were maintained in tissues for at least 120 days post CD4 depletion and even longer in circulation (at least 240 days post CD4^+^ cell depletion). Similar to CD8^+^ depletion studies, CD4^+^ T-cell depletion in the LNs took longer to reach completion than in any other tissue [139]. While CD4^+^ T-cell depletion persisted for a long period of time, the SMs did not progress to AIDS, even though at 18 months post depletion, the CD4^+^ T-cell counts were still lower than 100 cells/mm^3^. The CD4^+^ T-cell depletion had a significant impact on viral replication, though, with the VLs decreasing with 2–3 logs. There was also a strong correlation between the decrease in activated Ki67^+^ CD4^+^ T-cells and the decrease in VLs, and similarly between the increase in activated CD4^+^ T-cells and the rebound of VLs [139]. These results indicate that CD4^+^ T-cells primarily function as target cells for the virus, even in natural hosts.

Lastly, to assess the role of CD4^+^ T-cells in SIV disease progression and establish whether their preservation in natural hosts of SIV contributes to preventing intestinal dysfunction and protection from disease progression, a prolonged (>1 year) CD4^+^ T-cell depletion was experimentally induced in AGMs [306]. All circulating CD4^+^ T-cells and >90% of mucosal CD4^+^ T-cells were depleted for more than 18 months. As a result, plasma VLs and cell-associated viral RNA in tissues were lower in CD4^+^-cell-depleted animals than in controls. Meanwhile, CD4^+^-cell-depleted AGMs maintained gut integrity, controlled T-cell immune activation and inflammation, and did not progress to AIDS. Thus, in the absence of epithelial gut damage, persistent inflammation, and/or immune activation, CD4^+^ T-cell depletion alone is not sufficient to induce gut dysfunction or disease progression in natural hosts of SIV. Furthermore, their resistance to AIDS is independent of CD4^+^ T-cell counts and/or of a more swift and robust restoration of this cell subset in tissues [306].

#### 3.3.2. Depletion of Natural Killer Cells

NKs are effector lymphocytes of the innate immune system that help control several types of tumors and microbial infections by limiting their spread and subsequent tissue damage [307]. NK cells are also regulatory cells engaged in reciprocal interactions with dendritic cells, macrophages, T-cells, and endothelial cells. NK cells can therefore limit or exacerbate a variety of immune responses [308].

NKs have a critical role in bridging innate and adaptive immunity through the modulation of a complex cytokine network. Upon stimulation through NKG2D receptor signaling [309] or contact-dependent interactions with mature dendritic cells [310], NKs produce IFN-γ, thus promoting T-cell sensitization to the effects of IL-2 [311], T-helper 1 cell development, and macrophage activation. NKs are, however, highly plastic and, based on their environment, they can produce a variety of other type 1 (tumor necrosis factor [TNF-α/β], granulocyte–macrophage colony-stimulating factor [GM-CSF], macrophage inflammatory protein-α [MIP1α], IL-1, IL-8), type 2 (IL-3, IL-5, IL-10, IL-13), and type 3 (transforming growth factor-β [TGF-β]) cytokines, thus making a decisive contribution to the type 1/type 2 balance.

In addition to their immunoregulatory role, which is exerted through cytokine production, NKs also have an accessory function, which is exerted through cell-to-cell contact-dependent interactions with antigen-presenting cells (APCs). Such interactions can be both activatory and lytic [307]. Multiple APC surface molecules can be involved in such lytic interactions, either activatory, i.e., B7-1/B7-2, CD40L, CD54, HLA [310], or inhibitory, i.e., CD125 and MHC class I [307].

Two studies have reported that NK cell depletion during both acute and chronic SIVmac251 infection stages in RMs had no impact on virus replication. Both studies were based on the administration of an anti-human CD16 antibody, which is an FcγRIII receptor expressed only on NK cells in RMs. For the animals receiving anti-CD16 mAb during the chronic SIVmac infection, a single dose of 10 mg/kg was administered intravenously. Only 80–90% of the NKs from the peripheral blood were depleted, for only about 10 days, before returning to baseline levels. No changes in the plasma VLs were observed [312,313].

The impact of NKs on the control of viral replication during acute SIV infection was assessed by depleting NK cells (50 mg/kg of the anti-human CD16 antibody) one day prior to RM infection with SIVmac251. This depletion was about 90% effective and the NK cells returned to near baseline levels at 14 dpi. Even though the authors reported slightly higher median VLs in the NK-depleted group compared to controls, these differences did not reach statistical significance [312,313]. No change in the central memory populations (CD3^+^ CD4^+^ CD28^+^ CD95^+^) occurred followed NK depletion during acute SIV infection [312].

Not all NK cells from macaques express CD16. Therefore, while anti-CD16 mAb only targeted cytotoxic NK cells (CD16^high^ cells), it did not target all of the NK cells. Other NKs are CD16^low^ and exert their immune function via the secretion of chemokines and cytokines; these NK subsets were likely not affected by the in vivo depletion study, which may explain the lack of impact on viral replication. Moreover, the depletion was not sustained, which may have also prevented the identification of any impact on the pathogenesis. Finally, the efficacy of the NK depletion was only assessed in the periphery. In tissues, NK depletion was limited in the LNs and no information exists on the efficacy of NK depletion at mucosal sites. Finally, although this antibody targets NK cells, it may also deplete monocytes expressing CD16 [313].

In a second generation of NK depletion studies, NKs were targeted with a Janus kinase 3 (JAK3) inhibitor. The JAK3 inhibitor was administered daily at a 10 mg/kg dose for 35 days to both controller (low or undetectable plasma VLs) and progressor (high VLs) RMs. At the dose tested, the JAK3 inhibitor impacted both circulating and mucosal NK cells. NK cell depletion was associated with a transient increase in plasma and gastrointestinal (GI) tissue VLs [314]. While circulating NK cells eventually returned to baseline values, the GI-tract NKs remained persistently depleted [314]. However, these results have to be interpreted with prudence, as more recent findings have shown that the JAK3 inhibitor utilized in these studies has a broader activity than previously reported, with dose-dependent effects on both JAK2 and JAK1 [314]. This suggests that multiple pathways were likely affected with the administration of this drug (i.e., reduction in CD8^+^ T-cells and impact on dendritic cells) that need to be considered [315].

More recently, anti-IL-15 was reported to very effectively deplete NK cells in both blood and tissues [316]. This method was used to perform a pre-infection depletion of systemic NK cells, followed by a subsequent challenge with barcoded SIVmac239X. This NK depletion strategy was highly effective, nearly completely ablating all NK cell subsets in the blood, liver, oral, and rectal mucosa, as well as in LNs, and persisting throughout the study [317]. The frequencies and phenotypes of T-cells remained virtually unchanged, indicating minimal off-target effects. An early and sustained increase in viremia was observed (1–2 log increase over the controls). However, sequence analysis of the barcodes found no difference in the number of independent transmission events [317]. The depletions of total, central memory, and CCR5-expressing CD4^+^ T-cells were similar between the NK-depleted and control macaques. CD8^+^ T-cell activation was higher in the NK-cell-depleted RMs when measured according to Ki67 and PD-1 expression. A modest increase in the levels of inflammatory cytokines was observed in the NK-cell-depleted RMs compared to the controls. Collectively, these data suggest that NK cells are important modulators of HIV/SIV dissemination and pathogenesis, but they cannot independently eliminate individual transmission events [317].

In AGMs, NK cell depletion through treatment with anti-IL-15 mAb during chronic SIVagm infection resulted in high viral replication rates in the B-cell follicles and the T-cell zone and increased viral DNA in LNs [318]. As such, this experiment confirmed a proposed mechanism of the lack of pathogenicity of SIV infection in natural hosts, i.e., the early migration of NK cells into the LN follicles, where they contribute to limiting the viral reservoir by maintaining the follicles virus-free [318].

#### 3.3.3. Depletion of Regulatory T-Cells

Tregs suppress the cell-mediated immune response early in HIV/SIV infection [319]. In PWHs, the relative Treg frequency directly correlates with VLs and disease progression [166,320,321,322,323], and it is inversely correlated with SIV-specific CTL responses [319]. Acute CD4^+^ T-cell depletion partially spares Tregs, the fraction of which increases during this stage of infection [170,173,322,323]. In PWHs undergoing ART, Treg frequency returns to nearly baseline levels [165,323]. In circulation and in the rectal mucosa of the ECs, Treg frequencies are lower, though the absolute counts are higher, compared to progressors [320,324]. Tregs are relatively resistant to SIV-mediated cell death [325]. In progressors, the Treg suppressive capacity is maintained throughout infection [326], with enhanced function in the LNs [327,328,329].

Tregs are both beneficial and detrimental for HIV/SIV pathogenesis. They control the activation status of HIV-infected CD4^+^ T-cells [163,171,319,330,331]. In AGMs, Tregs increase as early as 24 h post SIV infection, concomitantly with TGF-β and IL-10 [332], in marked contrast with SIV-infected RMs, in which TGF-β increases are moderate and IL-10 increases are delayed [52]. CTLA-4 blockade early in SIV infection of RMs increased viral replication and decreased response to ART [333]. In vitro, Tregs can limit the infection of conventional CD4^+^ T-cells through DC-CD4^+^ T-cell immunological synapses [334]. The beneficial effects of Tregs on HIV/SIV pathogenesis are achieved through the control of activation status of virus-producing cells and their shift into the resting state, thereby suppressing viral production, which prevents the infection spreading. However, driving infected CD4^+^ T-cells into a resting state promotes HIV/SIV latency and thus reservoir seeding [335].

Treg frequency inversely correlates with SIV-specific CTLs, suggesting that Tregs suppress HIV-specific CD8^+^ T-cell responses and T-cell activation [173,319,336], and inhibit viral clearance. Treg depletion in humanized mice reduces the peak VLs and p24 intracellular staining in plasma and lymphoid tissues, such as the spleen and mesenteric LNs [337], supporting their negative effect on disease.

Tregs from PWHs retain their immunosuppressive activity [168,173,338]. HIV infection of Tregs results in a loss of functionality [332,339]. This altered Treg suppressive capacity may further the generalized immune activation in chronic infection.

Tregs also appear to be an important HIV reservoir, as (i) their frequency increases during HIV/SIV infection [165,166,170,173,319,320,321,322,323]; (ii) Tregs contain HIV/SIV DNA more frequently than non-Tregs [325,340]; (iii) Tregs have a better survival rate from SIV infection [325]; and (iv) SIV-infected Tregs exhibit impaired suppressive activity [332,339]. Together with their major role in shaping the viral reservoir, these data point to a key role for Tregs as targets for cure research strategies.

The suppressive function of Tregs during HIV infection has led to the idea that their in vivo ablation could be beneficial for PWHs. Yet, Tregs are also beneficial, as they suppress generalized immune activation; therefore, their depletion may have a deleterious effect on the pathogenesis of HIV/SIV infections. The major problem with depleting Tregs is that their marker, FoxP3, is intracellular, and therefore it cannot be directly targeted in vivo. Multiple other targets have, however, been considered for in vivo Treg depletion strategies.

**• Targeting CD25 for Treg depletion:** Two drugs target Treg through their constitutive expression of CD25 (the IL-2 receptor): Daclizumab and Ontak [341,342]. They have different mechanisms of action. Daclizumab is a monoclonal antibody to CD25 that prevents IL-2 from interacting with its receptor. It has been approved for the treatment of relapsing forms of multiple sclerosis [343,344] and in adult T-cell leukemia to induce remission [345]. For a short time, it was used to prevent acute graft rejection in patients with kidney transplants [346].

Ontak (Denileukin difitox) consists of IL-2 coupled with diphtheria toxin: IL-2 identifies and binds to the CD25^+^ cells, allowing the diphtheria toxin to enter the cell and cause cell death via the ADP-ribosylating host eEF-2 and the prevention of protein synthesis [347]. Ontak has been used to treat CD25^+^ cutaneous T-cell lymphoma [348], and has been tested with relatively positive results in peripheral T-cell lymphoma, metastatic renal cell carcinoma, and unresectable stage IV melanoma [349,350,351]. Very low doses of Ontak administrated to melanoma patients together with a dendritic cell vaccine produced no peripheral Treg depletion [352]. This limited efficacy is explained by the fact that Ontak internalization in activated Tregs occurs even at low concentrations, while Ontak internalization in resting Tregs requires very high concentrations [353]. This may be a potential barrier to the use of Ontak to target the resting reservoir.

Ontak depleted Tregs in the blood, spleen, and mesenteric LNs of DKO-hu HSC mice infected with HIV-R3A, and increased HLA-DR expression in CD4^+^ and CD8^+^ T-cells. However, the levels of HIV-1 in plasma and lymphoid organs were lower during acute HIV infection [337]. Meanwhile, Ontak administration to HIV-1-infected, ART-suppressed humanized NRG mice resulted in viral reactivation in the spleen and bone marrow. Cell-associated viral DNA levels did not change, indicating that virions relapsed from the reservoir. ART prevented the reactivated virus from reinfecting cells, and after virus control was achieved post Ontak administration, the levels of cell-associated viral DNA were significantly decreased in the lymphoid tissue compared to those observed in controls, with no significant change in total CD4^+^ T-cells in the spleen and bone marrow [354].

In chronically SIVsab-infected AGMs, Ontak determined a significant Treg depletion and induced significant CD4^+^ and CD8^+^ T-cell activation [161]. Administration during acute SIV infection was used to dissociate T-cell activation from inflammation. This intervention abolished the control of T-cell immune activation beyond the transition from acute to chronic infection, but had no effect on gut barrier integrity, microbial translocation, inflammation, or hypercoagulation. Persistent T-cell activation altered SIV pathogenesis in a natural host, shifting the ramp-up in viral replication to earlier time points, prolonging the high levels of replication, and delaying CD4^+^ T-cell restoration. Yet, there was no clinical or biological sign of disease progression in the Treg-depleted AGMs. Therefore, Treg depletion demonstrated that, in the absence of damages to mucosal barrier integrity, systemic T-cell activation alone is not sufficient to drive disease progression [355].

Finally, Ontak administration to SIVsab-infected RMs, a model of spontaneous complete control of HIV infection [236,356], resulted in the depletion of 75–85% of peripheral Tregs, an 8- to 10-fold increase in the activation of peripheral CD4^+^ and CD8^+^ T-cells, a boost of SIV-specific T-cells, and a relatively robust virus reactivation [357], suggesting that Treg depletion is a plausible strategy for reducing the HIV reservoir in circulation and lymphoid tissues, while boosting cell-mediated immune responses [357].

Ontak is currently discontinued. A new bivalent IL-2 immunotoxin with increased potency when compared to the Ontak-like monovalent version has been developed [358]. In a human CD25^+^ HUT102/6TG tumor-bearing NSG mouse model, this bivalent immunotoxin significantly prolonged the mice’s survival in a dose-dependent manner [352]. In RMs, this treatment resulted in significant depletion of the circulating Tregs (>70%) and their partial depletion in the gut (25%) and LNs (>50%) [359]. The fractions of CD4^+^ T-cells expressing Ki-67 increased up to 80%, paralleled by increases in the inflammatory cytokines. In the absence of ART, the plasma virus rebounded to 10^3^ vRNA copies/mL by day 10 after IL-2-DT administration [359]. A large but transient boost of the SIV-specific CD8^+^ T-cell responses occurred. However, when IL-2-DT was given to SIVmac-infected RMs on ART, treatment had to be discontinued because of high toxicity and lymphopenia [359]. As such, while all treatments significantly depleted Tregs, the side effects in the presence of ART precluded their clinical use, calling for different Treg depletion approaches [359].

**• Targeting CCR4 for Treg depletion:** Tregs express a high level of CCR4 [360,361,362], the receptor for CC chemokines (MIP-1, RANTES, TARC, and MCP-1), which is also a coreceptor for HIV-1 [363]. A diphtheria-toxin-based anti-human CCR4 immunotoxin has been developed that can cause protein synthesis inhibition in target cells upon binding. It prolonged the survival of tumor-bearing NOD/SCID IL-2 receptor γ−/− (NSG) mice injected with human CCR4^+^ acute lymphoblastic leukemia cells [364]. In RMs, it depleted ~80% of CCR4^+^ FoxP3^+^ and 40% of FoxP3^+^ CD4^+^ circulating T-cells. In the LNs, although there was a decrease of ~90% in the CCR4^+^ FoxP3^+^ Tregs, overall, FoxP3^+^ CD4^+^ T-cells were decreased by only 9–22% [365]. This drug was less effective compared to IL-2-DT in activating T-cells and reactivating the virus in chronically SIV-infected RMs [359]. An anti-CCR4 monoclonal antibody, Mogamulizumab is used for treating peripheral T-cell lymphoma and cutaneous T-cell lymphomas like mycosis fungoides and Sezary syndrome, and acts by depleting CCR4^+^ malignant cells and CCR4^+^ Tregs [366,367].

**• Cyclophosphamide (Cy):** Cyclophosphamide (Cy) is an alkylant agent used for the treatment of leukemias, lymphomas, allogeneic stem-cell transplantation [368], and systemic lupus erythematosus (SLE) [369,370,371]. In low, metronomic dosages, Cy can be used to selectively and significantly deplete and reduce Treg functionality [372,373,374]. Treg selectivity has been attributed to decreased DNA repair, a decreased production of glutathione, and decreased ATP levels of Tregs, which abrogate glutathione production, thereby inducing hypersensitivity to Cy [375]. Proliferating and activated CCR2^+^ Tregs are targeted by Cy [376].

In an HIV-positive patient with SLE, treatment with Cy induced an enormous burst in viral replication (of up to >10^7^ copies/mL) [377]. Using escalating doses, Cy had little effect on HIV DNA in LNs and PBMCs and plasma VLs, and there was no significant difference in the HIV DNA burden of LNs and PBMCs versus the control group; note that VLs were not suppressed in these patients, with two-fifths of patients admitting to nonadherence to ART [378]. Thus, it is possible that the increase in plasma VLs and lack of viral DNA clearance may have been due to nonadherence.

Low and high doses of Cy were administered to SIV-infected RMs to assess their effects on the SIV reservoirs. As a Treg-depleting agent, Cy unselectively depleted Treg and total lymphocytes, resulting in minimal immune activation and no viral reactivation [379]. As a cytoreductive agent, Cy induced massive viral reactivation in EC RMs without ART [379]. However, when administered with ART, Cy had substantial adverse effects, including mortality [379]. As such, Cy is likely not suitable as an HIV cure agent.

**• Interventions targeting Treg function:** Various therapies affecting Treg function have also been tested. CTLA-4^+^PD-1^neg^ CD4^+^ T-cells from multiple tissues are enriched for replication-competent SIV in infected RMs treated with ART, suggesting a potential therapeutic target for reservoir elimination [380]. During HIV infection, CTLA-4 contributes to HIV-specific T-cell suppression. Inversely, CTLA-4 blockade enhances CD4^+^ T-cell functionality, i.e., IFN-γ production and cell proliferation [381,382]. In a PWH treated with Ipilimumab (α-CTLA-4 mAb) for melanoma, VLs remained below the limit of detection of standard techniques. However, a general decline in VLs was seen when using the single copy assay; post treatment, there was an increase in cell-associated unspliced RNA, likely due to the expansion of infected T-cells [383]. Additionally, CTLA-4 blockade in RMs resulted in decreases in viral RNA, and increases in the SIV-specific immune responses [384]. However, when used in early infection with a pathogenic SIVmac251-infected RM model, the CTLA-4 blockade had opposite effects, increasing immune activation, viral replication, and IDO; having virtually no impact on responses to vaccination, i.e., SIV-specific responses; and abrogating the response to ART [333].

Immune checkpoint blockade using monoclonal antibodies for PD-1, CTLA-4, and dual CTLA-4/PD-1 was tested in SIV-infected, ART-suppressed RMs [385]. Dual blockade was substantially more effective than PD-1 blockade alone, enhancing T-cell cycling and differentiation, expanding effector memory T-cells, and inducing robust viral reactivation in plasma and PBMCs [385]. Yet, none of these interventions enhanced SIV-specific CD8^+^ T-cell responses during ART or viral control after ART interruption. Therefore, it was concluded that CTLA-4/PD-1 blockades were still insufficient for reservoir clearance and achieving viral control, indicating that immune checkpoint blockade regimens are unlikely to induce HIV remission in the absence of additional interventions [385].

The administration of IDO inhibitor 1-methyl-D-tryptophan (D-1mT) combined with CTLA-4 blockade to SIVmac251-infected macaques on ART did not change their VLs. This treatment also induced acute pancreatitis, showcasing that Tregs should be manipulated in vivo with caution [386].

**Table 1 viruses-16-00972-t001:** Reagents used for the in vivo depletion of different populations of immune cells: usage, dosage, and efficacy.

Antibody	Description	Target Cell	Dosage ^1^	Duration of Depletion (Blood)	Effects on Other Tissues	References
CD8β255R1	Chimeric mouse-rhesus mAb	CD8αβ^+^	50 mg/kg by i.v.	16 weeks	Depletes specifically CD8^+^ T-cells in blood and LNs (≈50%); minimal impact on NK cells and γδ^+^ cells	[224]
M-T807R1	Chimeric mouse-rhesus mAb	CD8α^+^	Day 0: 50 mg/kg by i.v., Days 6 and 13: 10 mg/kg by i.v.	3–5 weeks	Depletes CD8^+^ cells in LNs, the intestine, and the genital tract; downregulates CD8 in the gut	[23,118,182,204,205,206,207]
cM-T807	Chimeric mouse–human mAb	CD8α^+^	Day 0: 50 mg/kg by i.v., Days 6 and 13: 10 mg/kg by i.v.	2–5 weeks	Depletes CD8^+^ cells in LNs, the intestine, and the genital tract; downregulates CD8 in the gut	[118,132,179,202,204,214,215,217,226,236]
OKT8F	Chimeric mouse–human mAb	CD8α^+^	1 mg/kg for 4 successive days; 4 mg/kg for 3 successive days	<2 weeks	Depletes CD8^+^ cells in LNs and bone marrow	[115,129,198,199,387]
TRX2	Chimeric mouse–human mAb	CD8α^+^	Days 0, 1, and 3: 3 mg/kg; Days 6, 10, and 13: 6 mg/kg; Days 17 and 20: 9 mg/kg by i.v.	7 weeks	Depletes CD8^+^ cells in LNs and the spleen	[209]
T87PT3F9	Chimeric mouse–human mAb	CD8α^+^	Day 0 and 7: 2 mg/kg by i.v.	2 weeks	Depletes CD8^+^ cells in LNs	[208]
Rituximab	Chimeric mouse–human mAb	CD20^+^	50 mg/kg every 3 weeks up to 200 days pi by i.v.	300 days	Depletes CD20^+^ cells in blood, LNs, and the intestine; efficacy in suppressing humoral response is predicted by the degree of depletion in tissues	[283,284,285,286,288]
CD4R1	Chimeric mouse-rhesus mAb	CD4	50 mg/kg by i.v. every three weeks (up to 21 administrations)	>80 weeks	Massive depletion in blood and lymph nodes; partial depletion in the gut	[140,141,306]
OKT4A-huIgG1	Humanized mAb	CD4^+^	10 mg/kg i.v., then 3 × 5 mg/kg every 4 days (or 4 × 10 mg/kg i.v., every 4 days)	240 days dpi	Depletes CD4^+^ efficiently in LNs and bone marrow; is not as effective in mucosal tissues	[139]
3G8	Chimeric mouse–human mAb	CD16^+^	50 mg/kg by i.v.	Transient depletion	Unknown efficacy in tissues other than blood	[312,313]
CP-690550	JAK3 inhibitor	NKs	Loading dose of 20 mg/kg, then 10 mg/kg daily, 35 days, p.o.	6 weeks in periphery, longer at mucosal sites	Depletes both circulating and mucosal NKs; broad activity (impacts also JAK2 and JAK1)	[314]
SM17-25	Anti-IL-15 mAb	NKs	20 mg/kg, then 10 mg/kg in one-week intervals	2–3 weeks	Depletes both circulating and tissue NKs	[316,317,318]
MDX-010	Anti-CTLA-4 mAb	Blocks CTLA-4^+^ function	4 × 10 mg/kg every 3 weeks	Not assessed; effects tested at 14 weeks dpi	Impacts rectal mucosa more than LNs	[333]
Ontak	Fused diphtheria toxin moiety with IL-2	CD4^+^ CD25^+^Tregs	3 × 15 mg/kg every 3 weeks, for 5 consecutive days	2–3 weeks	Depletes CD4^+^ CD25^+^ blood; is less effective in the LNs and intestine	[161,355,357]
Anti-IL-2-DT	Bivalent diphtheria toxin moiety with IL-2	CD4^+^ CD25^+^Tregs	25 mg/kg B.I.D. for five consecutive days i.v.	2–3 weeks	Depletes CD4^+^ CD25^+^ blood; is less effective in the LNs and intestine	[359]
Anti-CCR4-DT	Bivalent diphtheria toxin moiety with CCR4	CD4^+^ CCR4^+^Tregs	25 μg/kg, twice a day (BID) for 5 days	2–3 weeks	Modest and transient Treg depletion	[359]

^1^ Only the most effective dosage is listed; i.v.—intravenously; p.o.—per os; pi—post infection.

## 4. Conclusions

The in vivo depletion of selected components of the immune system has provided answers to many important questions of HIV pathogenesis. While such studies are not a perfect science and can have many confounding variables, they are our current best option to establish the contribution of different immune effectors to HIV pathogenesis. Specifically, depletion studies have shown the importance of CD8^+^ T-cells in controlling virus replication during both acute and chronic SIV infections [118,202].

After the failure of clinical HIV vaccine trials, there is a large consensus that successful HIV vaccines should be ones that elicit CD8^+^ and CD4^+^ T-cell responses, as well as NAb responses [214]. The ideal situation would be to find a vaccine that elicits a high-titer antibody response and one that displays strong neutralization abilities against HIV-1. However, the key word is “ideal”, indicating that this might not be a possibility in the near future of HIV research. Therefore, further depletion studies in animals utilizing improved mAb technology or other methods of depletion (i.e., CAR-T) in the future may be critical for determining the primary correlates of immune protection in HIV infection.

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
