# Peer review of "Making a Monkey out of Human Immunodeficiency Virus/Simian Immunodeficiency Virus Pathogenesis: Immune Cell Depletion Experiments as a Tool to Understand the Immune Correlates of Protection and Pathogenicity in HIV Infection"

_viruses, 2024, doi:10.3390/v16060972_

Round 1
Reviewer 1 Report
Comments and Suggestions for Authors
This article provides a comprehensive review of using in vivo depletion of select immune cell subsets to define the immune correlates of infection in nonhuman primate models of HIV infection. The authors extensively cover humoral and cellular immune responses during HIV/SIV infections. They also present a detailed overview of depletion studies conducted in natural and non-natural hosts of SIV infections, highlighting their significance and limitations. However, the manuscript lacks clarity and focus. It will benefit from editing to make it more concise and correct statements that are either unclear or incorrect.
Major points:
1. The introduction needs to be more concise and focused, clearly stating the main objective and scope of the review.
2. The section describing the immune responses during HIV/SIV infection is extensive, but it could be better organized and streamlined to improve the flow and clarity of the information presented.
3. Discussion of the various T cell depletion studies is overwhelming and lacks a narrative putting the studies into context. A table summarizing the various studies and highlighting key points may be beneficial.
4. There are instances where redundant information is presented in back-to-back sentences or even within sentences. For example, point (i) in lines 373-375 and lines 413-416.
5. When introducing SHIV models (lines 57-69), the authors fail to mention a notable advantage of these chimeric viruses: most of them contain HIV envelopes, making them valuable tools for evaluating anti-HIV antibodies in vivo.
6. Imprecise language is found throughout the manuscript. For example, line 93: …mandrills (Mandrillus sphinx) are massively infected with species-specific viruses… It is unclear what "massively infected" means in this context. It would be more appropriate to state, "…mandrills (Mandrillus sphinx) are naturally infected with species-specific viruses, but they generally do not progress to AIDS".
7. It is difficult to follow the logic in section 2.1. Contradictory statements are made in lines 146-158. The first paragraph in this section states that NAbs are ineffective, but with no transition, the second paragraph describes how NAbs work (lines 151-158).
8. The description of ADCC in lines 162-164 is wrong. ADCC is not the engulfment of antibody covered cells by phagocytes, which is antibody-dependent phagocytosis (ADCP), but rather the triggered release of cytotoxic molecules by NK cells after the engagement of their Fc receptors by antigen-bound antibodies. This paragraph then describes antibody-dependent complement deposition (ADCD) and activation without establishing how complement mediates virolysis. This paragraph needs to be rewritten to describe ADCC accurately, and the discussion of complement should be moved to the following paragraph.
9. Line 184: This sentence should not start with "In theory" because it is well established that the classical pathway of complement activation can lead to the lysis of pathogens or infected cells. It is also worth noting that HIV incorporates host complement control proteins into its envelope, inhibiting complement-mediated virolysis.
10. Lines 222-223: Cellular immunity is mediated by various cells, including those listed. The authors should clarify that Th17 and Tregs are subsets of CD4 T cells.
11. Line 242: The authors should introduce to the reader that Mamu-A*01 is an MHC class I molecule.
12. Lines 344-346 are confusing as written.
13. Lines 546-547: It is inaccurate to state that HIV vaccines have largely focused on eliciting CTL responses. This was true briefly in the mid-2000s, but the focus has returned chiefly to designing vaccines that elicit neutralizing antibodies.
14. Lines 568-570: This sentence should be removed, or additional information on the study outcome and its significance should be included.
15. Lines 686-692 have limited value because an antibody that can distinguish CD8β from CD8α is now available.
16. Section 3.1.7 needs a topic sentence to introduce the concepts discussed in this section.
17. Section 3.1.8: Mamu-B*08 should also be included as a protective MHC class I allele in RMs.
Minor points:
1. Line 29: Should it be "Although each of these methods…" instead of "Although every of these methods…"?
2. Lines 384-386: What approaches were used and their limitations?
3. Lines 413-416: These sentences repeat the same information.
4. Line 537: predepletion is misspelled.
5. Line 565-566: "…, in contrast to CD8+-cell depletion studies in SIVmac239∆nef-infected RMs" can be removed from this sentence for clarity.
6. Lines 824-826: This sentence is unclear.
7. Line 861: Clarify that antibody-based vaccines were the first choice for HIV vaccine development.
8. Line 942: Remove extraneous "of".
Comments on the Quality of English Language
Author Response
Reviewer 1
This article provides a comprehensive review of using in vivo depletion of select immune cell subsets to define the immune correlates of infection in nonhuman primate models of HIV infection. The authors extensively cover humoral and cellular immune responses during HIV/SIV infections. They also present a detailed overview of depletion studies conducted in natural and non-natural hosts of SIV infections, highlighting their significance and limitations. However, the manuscript lacks clarity and focus. It will benefit from editing to make it more concise and correct statements that are either unclear or incorrect.
Thank you very much for your suggestions. We eliminated redundancies and shortened the manuscript. We also added a synopsis table, as indicated
Major points:
- The introduction needs to be more concise and focused, clearly stating the main objective and scope of the review.
We refocused the introduction, as requested.
- The section describing the immune responses during HIV/SIV infection is extensive, but it could be better organized and streamlined to improve the flow and clarity of the information presented.
We streamlined this section
- Discussion of the various T cell depletion studies is overwhelming and lacks a narrative putting the studies into context. A table summarizing the various studies and highlighting key points may be beneficial.
We added a Table1, as suggested
- There are instances where redundant information is presented in back-to-back sentences or even within sentences. For example, point (i) in lines 373-375 and lines 413-416.
We eliminated redundancies
- When introducing SHIV models (lines 57-69), the authors fail to mention a notable advantage of these chimeric viruses: most of them contain HIV envelopes, making them valuable tools for evaluating anti-HIV antibodies in vivo.
We added this, as suggested
- Imprecise language is found throughout the manuscript. For example, line 93: …mandrills (Mandrillus sphinx) are massively infected with species-specific viruses… It is unclear what "massively infected" means in this context. It would be more appropriate to state, "…mandrills (Mandrillus sphinx) are naturally infected with species-specific viruses, but they generally do not progress to AIDS".
We corrected this explaining that there are tremendous levels of prevalence of infection in natural hosts
- It is difficult to follow the logic in section 2.1. Contradictory statements are made in lines 146-158. The first paragraph in this section states that NAbs are ineffective, but with no transition, the second paragraph describes how NAbs work (lines 151-158).
We corrected the contradictory statements
- The description of ADCC in lines 162-164 is wrong. ADCC is not the engulfment of antibody covered cells by phagocytes, which is antibody-dependent phagocytosis (ADCP), but rather the triggered release of cytotoxic molecules by NK cells after the engagement of their Fc receptors by antigen-bound antibodies. This paragraph then describes antibody-dependent complement deposition (ADCD) and activation without establishing how complement mediates virolysis. This paragraph needs to be rewritten to describe ADCC accurately, and the discussion of complement should be moved to the following paragraph.
We completely replaced this paragraph, as suggested
- Line 184: This sentence should not start with "In theory" because it is well established that the classical pathway of complement activation can lead to the lysis of pathogens or infected cells. It is also worth noting that HIV incorporates host complement control proteins into its envelope, inhibiting complement-mediated virolysis.
We incorporated this suggestion in the revised version
- Lines 222-223: Cellular immunity is mediated by various cells, including those listed. The authors should clarify that Th17 and Tregs are subsets of CD4 T cells.
Modified as requested
- Line 242: The authors should introduce to the reader that Mamu-A*01 is an MHC class I molecule.
Done as requested
- Lines 344-346 are confusing as written
Corrected as requested
- Lines 546-547: It is inaccurate to state that HIV vaccines have largely focused on eliciting CTL responses. This was true briefly in the mid-2000s, but the focus has returned chiefly to designing vaccines that elicit neutralizing antibodies.
Modified, as requested
- Lines 568-570: This sentence should be removed, or additional information on the study outcome and its significance should be included.
Done
- Lines 686-692 have limited value because an antibody that can distinguish CD8βfrom CD8α is now available.
Deleted, as suggested
- Section 3.1.7 needs a topic sentence to introduce the concepts discussed in this section.
Done, as requested
- Section 3.1.8: Mamu-B*08 should also be included as a protective MHC class I allele in RMs.
Done
Minor points:
- Line 29: Should it be "Although each of these methods…" instead of "Although every of these methods…"?
Yes. Modified
- Lines 384-386: What approaches were used and their limitations?
Corrected
- Lines 413-416: These sentences repeat the same information.
Corrected
- Line 537: predepletion is misspelled
Corrected
- Line 565-566: "…, in contrast to CD8+-cell depletion studies in SIVmac239∆nef-infected RMs" can be removed from this sentence for clarity.
Done
- Lines 824-826: This sentence is unclear.
Modified
- Line 861: Clarify that antibody-based vaccines were the first choice for HIV vaccine development.
Clarified
- Line 942: Remove extraneous "of".
Removed

Reviewer 2 Report
Comments and Suggestions for Authors
This is a very comprehensive and exceptional review of nonhuman primates studies to understand HIV/SIV pathogenesis.
There is a brief introduction of various NHP models to study HIV pathogenesis, followed by a thorough description of innate and adaptive immune responses to infection. To understand the relative contribution of various immune responses to pathogenesis, several studies have been conducted in the past that depleted specific immune cell populations using rhesusized antibodies. The authors go in great depth to discuss the outcome and biological significance of these studies depending on, but not limited to, the NHP model, antibody clone usage, stage of infection, and antiretroviral treatment. The review will serve as a tremendous resource for many researchers.
Author Response
This is a very comprehensive and exceptional review of nonhuman primates studies to understand HIV/SIV pathogenesis.
There is a brief introduction of various NHP models to study HIV pathogenesis, followed by a thorough description of innate and adaptive immune responses to infection. To understand the relative contribution of various immune responses to pathogenesis, several studies have been conducted in the past that depleted specific immune cell populations using rhesusized antibodies. The authors go in great depth to discuss the outcome and biological significance of these studies depending on, but not limited to, the NHP model, antibody clone usage, stage of infection, and antiretroviral treatment. The review will serve as a tremendous resource for many researchers.
Thank you very much for your appreciation
Reviewer 3 Report
Comments and Suggestions for Authors
Overall: Jen Symmonds et al. evaluate the impact of immune cell depletion majority in non-human primate model. The authors explore and discuss about the result observed and the limit associated. With non-exhaustive references and well-developed manuscript, the study is great quality, some details are commented just below to finalize the manuscript.
Comments:
- Line 156: " 'mature" " instead of " "mature" "
- Line 194 to 195: it is important to mention here that there is a dysfunction of IgA-producing from B cells and plasma cells during HIV/SIV infection. Thus, it is important to note that the loss of IgA in plasma correlates with and intestine during HIV/SIV infection. While the hypergammaglobulinemia is associated with chronic SIV pathological infection but not in natural host (10.3389/fimmu.2017.01581, 10.1371/journal.ppat.1006087, 10.1038/s42003-022-03619-y, 10.1128/JVI.01887-13).
- Line 220: Chapter 2.2 is "cellular T immune response" not "cellular immune response"
- Chapter 2.5: The authors should also mention that the contribution of inducible regulatory T cells (iTregs) in HIV/SIV infection remains largely unknown.
- Line 440: A space is missing "4to"
- In section 3.3.1. you should mention that the CD4+ T cell depletion studies have shown that do not impact the other cell types such as monocytes, dendritic cell.
Author Response
Overall: Jen Symmonds et al. evaluate the impact of immune cell depletion majority in non-human primate model. The authors explore and discuss about the result observed and the limit associated. With non-exhaustive references and well-developed manuscript, the study is great quality, some details are commented just below to finalize the manuscript.
Thank you very much for your appreciation
Comments:
- Line 156: " 'mature" " instead of " "mature" "
Modified as requested
- Line 194 to 195: it is important to mention here that there is a dysfunction of IgA-producing from B cells and plasma cells during HIV/SIV infection. Thus, it is important to note that the loss of IgA in plasma correlates with and intestine during HIV/SIV infection. While the hypergammaglobulinemia is associated with chronic SIV pathological infection but not in natural host (10.3389/fimmu.2017.01581, 10.1371/journal.ppat.1006087, 10.1038/s42003-022-03619-y, 10.1128/JVI.01887-13).
Added, as requested
- Line 220: Chapter 2.2 is "cellular T immune response" not "cellular immune response"
Modified as per reviewer’s request
- Chapter 2.5: The authors should also mention that the contribution of inducible regulatory T cells (iTregs) in HIV/SIV infection remains largely unknown.
We added this discussion
- Line 440: A space is missing "4to"
Added
- In section 3.3.1. you should mention that the CD4+ T cell depletion studies have shown that do not impact the other cell types such as monocytes, dendritic cell.
Added, as requested
Round 2
Reviewer 1 Report
Comments and Suggestions for Authors
This article provides a thorough overview of in vivo immune subset depletion studies, highlighting key findings that enhance our understanding of the immune correlates of HIV/SIV protection and pathogenicity. The revisions have improved the manuscript's readability, and the table provides a convenient overview of the significant studies discussed. However, a few points remain unclear or imprecise. Clarifying these items will improve the manuscript before publication.
Major points:
1. Suggest revising the text in the abstract and removing colloquial language. Suggested edits:
a. Line 21-22: simplify the first line, “Understanding the underlying mechanisms of HIV pathogenesis is crucial for designing successful HIV vaccines and cure strategies.”
b. In lines 22-23: change “…this is a complicated matter, as the virus directly interacts with key players in the immune system,…” to “However, achieving this goal is complicated by the virus directly interacting with immune cells,…”
c. Simplify the text in line 26: “…, making it difficult to untangle the various concurrent mechanisms of HIV pathogenesis.”
2. Line 66: briefly describe transmitted-founder viruses or remove this sentence.
3. Lines 106-114: This paragraph is difficult to follow; suggest simplifying the text. For example, the first sentence could be revised to “Finally, multiple approaches are being developed to study HIV in animal models:…”
4. Lines 137-139: This sentence is confusing, suggest revising it to “However, the heavy glycosylation of HIV and SIV gp120 makes it difficult to elicit NAbs”.
5. The ADCVI assay was developed in the 2000s to assess the impact of non-neutralizing antibodies on virus replication. However, the term “ADCVI” is not regularly used to describe the collection of non-neutralizing antibody activity. To avoid confusion, the section starting with line 143 should discuss “Fc-mediated antibody effector functions”, which is a more appropriate umbrella term for discussing non-neutralizing antibody activity: ADCC, ADCP, and antibody-dependent complement deposition.
6. Lines 155-156: This sentence is confusing. Should it be the later assay that measures the inhibition of the virus by antibodies?
7. Lines 214-216: In the mid-2000s, there was significant interest in developing CD8 T-cell-based vaccines, but it may be too strong to say that there was “a relative consensus in the field” to do so. Suggest revising this sentence to note that the variability of HIV Env, the difficulty inducing bNAbs, and the potential to attack conserved regions of HIV made CD8+ T-cell-based vaccines attractive.
Minor points:
1. Line 81: A word appears to be missing from this sentence.
2. Line 130: Suggest replacing “whipped out” with “eliminated.”
3. Lines 194-196: This sentence contains redundant information provided in the previous sentence; revise for clarity.
4. Lines 303-303: This line is confusing, is it missing a word?
5. Line 370: Define MAb since this term is introduced here.
6. Line 772: “of” is misspelled as “fo”.
7. Line 1025: “leading” is misspelled as “lading.”
8. Line 1213: there is an extra “d” is in “bind.”
Comments on the Quality of English Language
Author Response
This article provides a thorough overview of in vivo immune subset depletion studies, highlighting key findings that enhance our understanding of the immune correlates of HIV/SIV protection and pathogenicity. The revisions have improved the manuscript's readability, and the table provides a convenient overview of the significant studies discussed. However, a few points remain unclear or imprecise. Clarifying these items will improve the manuscript before publication.
Thank you very much for your help to improve the quality of our work. We operated all the requested corrections.
Major points:
- Suggest revising the text in the abstract and removing colloquial language. Suggested edits:
- Line 21-22: simplify the first line, “Understanding the underlying mechanisms of HIV pathogenesis is crucial for designing successful HIV vaccines and cure strategies.”
Modified as requested. Replaced “crucial” in the suggestion with “critical”.
- In lines 22-23: change “…this is a complicated matter, as the virus directly interacts with key players in the immune system,…” to “However, achieving this goal is complicated by the virus directly interacting with immune cells,…”
Modified as requested
- Simplify the text in line 26: “…, making it difficult to untangle the various concurrent mechanisms of HIV pathogenesis.”
Modified as requested.
- Line 66: briefly describe transmitted-founder viruses or remove this sentence.
Modified as requested
- Lines 106-114: This paragraph is difficult to follow; suggest simplifying the text. For example, the first sentence could be revised to “Finally, multiple approaches are being developed to study HIV in animal models:…”
Changed the first sentence as suggested and streamlined the remaining paragraph.
- Lines 137-139: This sentence is confusing, suggest revising it to “However, the heavy glycosylation of HIV and SIV gp120 makes it difficult to elicit NAbs”.
Modified as requested
- The ADCVI assay was developed in the 2000s to assess the impact of non-neutralizing antibodies on virus replication. However, the term “ADCVI” is not regularly used to describe the collection of non-neutralizing antibody activity. To avoid confusion, the section starting with line 143 should discuss “Fc-mediated antibody effector functions”, which is a more appropriate umbrella term for discussing non-neutralizing antibody activity: ADCC, ADCP, and antibody-dependent complement deposition.
Modified as requested
- Lines 155-156: This sentence is confusing. Should it be the later assay that measures the inhibition of the virus by antibodies?
Sorry for the confusion. We modified, as suggested.
- Lines 214-216: In the mid-2000s, there was significant interest in developing CD8 T-cell-based vaccines, but it may be too strong to say that there was “a relative consensus in the field” to do so. Suggest revising this sentence to note that the variability of HIV Env, the difficulty inducing bNAbs, and the potential to attack conserved regions of HIV made CD8+ T-cell-based vaccines attractive.
Thank you for the suggestion, we modified as requested.
Minor points:
- Line 81: A word appears to be missing from this sentence.
Word added.
- Line 130: Suggest replacing “whipped out” with “eliminated.”
Modified
- Lines 194-196: This sentence contains redundant information provided in the previous sentence; revise for clarity.
Modified as requested
- Lines 303-303: This line is confusing, is it missing a word?
Added the missing word
- Line 370: Define MAb since this term is introduced here.
Modified as requested
- Line 772: “of” is misspelled as “fo”.
Modified as requested
- Line 1025: “leading” is misspelled as “lading.”
Modified as requested
- Line 1213: there is an extra “d” is in “bind.”
Corrected. Thank you very much for these corrections.
